# EIP: Weighted Ranking of LLMs by Quantifying Question Difficulty

**Xingjian Hu**[1], **Ziqian Zhang**[1,*] **Yue Huang**[2], **Kai Zhang**[1], **Ruoxi Chen**[3], **Yixin Liu**[1], **Qingsong Wen**[4], **Kaidi Xu**[5], **Xiangliang Zhang**[2], **Neil Zhenqiang Gong**[6], **Lichao Sun**[1,†]

[1]Lehigh University, [2]University of Notre Dame, [3]Zhejiang Wanli University,
[4]Squirrel Ai Learning, [5]City University of Hong Kong, [6]Duke University

## Abstract

Benchmarks establish a standardized evaluation framework to systematically assess the performance of large language models (LLMs), facilitating objective comparisons and driving advancements in the field. However, existing benchmarks fail to differentiate question difficulty, limiting their ability to effectively distinguish models' capabilities. To address this limitation, we propose Empirical Interaction Propagation (EIP), a novel framework designed to quantify both question difficulty and model competency. EIP introduces difficulty as the primary criterion for differentiation, enabling a more fine-grained evaluation of LLM capabilities. EIP's core mechanism facilitates bidirectional score propagation between models and questions. The core intuition of EIP is that a model earns a competency score when it correctly answers a question, while a question's difficulty score increases when it challenges a model. Using this framework, we evaluate 30 models on 35,550 questions across multiple domains. EIP achieves 90% agreement with human judgments and consistently outperforms strong baselines such as IRT. It also exhibits strong stability, fast convergence, and high computational efficiency, making it a practical solution for large-scale, difficulty-aware LLM evaluation. Code is available at https://github.com/Leozz04/EIP.

## 1 Introduction

Large Language Models (LLMs) have shown impressive breadth across tasks from natural language understanding to multi-step problem solving (Young et al., 2018; Thirunavukarasu et al., 2023; Hadi et al., 2023; Kaddour et al., 2023; Zhou et al., 2023). As capability grows, reliable evaluation becomes the community's dashboard. However, popular benchmarks—e.g., MMLU-Pro (Wang et al., 2024) and MATH (Hendrycks et al., 2021b)—typically collapse performance to *accuracy within topical categories*, implicitly treating all items as equally informative. This masks differences that hinge on *difficulty* and can flip model rankings when the mixture of easy vs. hard items shifts. Counting a single arithmetic fact and a multi-step calculus derivation as equally "correct" illustrates the problem: the evaluation fails to distinguish routine pattern matching from advanced reasoning.

We posit that difficulty must be modeled explicitly at the item level. We introduce **Empirical Interaction Propagation (EIP)**, a simple, robust, and scalable framework that jointly estimates *question difficulty* and *model competency* from observed successes/failures. Inspired by PageRank(Page et al., 1999), EIP constructs a directed bipartite interaction graph between models and questions and performs **damped bidirectional score propagation**: solving a hard question carries more evidential weight for competency; failing an easy question contributes more weight to difficulty. The process converges to a unique stationary solution, yielding difficulty and competency scores that support difficulty-aware comparison at scale. Compared to Item Response Theory (IRT), EIP is *non-parametric*, avoids per-item logistic fits, and runs in *linear time* in the number of model–question interactions, making it practical when per-model sample sizes are small and datasets are large.

---

*Equal contribution, random order.
†Corresponding author.

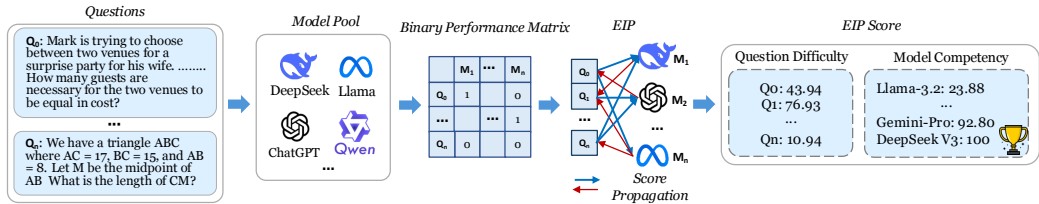

Figure 1: Schematic of EIP's weighted ranking pipeline.

**What EIP brings (methodological advantages).** We highlight five core advantages that make EIP practical and reliable for today's evaluation regimes: 1) *Human alignment*: difficulty estimates reach 90% consensus with human judgments, outperforming multiple IRT baselines across datasets; 2) *Efficiency and scalability*: propagation converges in 0.006 s on consumer hardware and scales linearly with the number of model–question interactions, supporting tens of thousands of items and dozens of models; 3) *Robustness*: question-side difficulty rankings remain highly stable under removal of individual models ($\rho = 0.998$), while model-side competency rankings remain stable under dataset perturbations ($\rho > 0.99$); 4) *Sensitivity beyond accuracy*: in controlled simulations, EIP recovers subtle performance gaps missed by accuracy- and IRT-based baselines, rewarding performance on harder items and enabling meaningful re-ordering among close models (adjacent changes consistent with Kendall's $\tau = 0.8492$); 5) *Simplicity and reproducibility*: a non-parametric graph procedure with a single damping hyperparameter; no per-item calibration or curve fitting is required, enabling easy, consistent deployments.

**What EIP reveals (empirical findings).** Applying EIP at scale surfaces actionable empirical patterns: 1) *Dataset-specific difficulty profiles*: MATH and MMLU-Pro exhibit broader difficulty tails suited for advanced reasoning, whereas GSM8k and HellaSwag skew easier; 2) *Model-family consistency*: families preserve characteristic difficulty patterns across parameter scales—scaling primarily shifts absolute accuracy rather than the relative difficulty structure; 3) *Open-weight model potential*: difficulty estimated on open-weight pools closely tracks full-pool results (Spearman 0.96, Pearson 0.94, Kendall 0.85), rivaling proprietary pools; 4) *Diversity benefits*: mixed-scale model pools reduce extreme misestimation by 83% relative to homogeneous pools and best align with human judgments (90% consensus); 5) *Size–performance correlation*: larger and proprietary models dominate the top competency scores, consistent with observed scaling trends.

We evaluate EIP across **30 models** and **35,550 questions** spanning BBH, GPQA, GSM8k, HellaSwag, MATH, and MMLU-Pro. Our study covers human alignment, convergence, robustness under model/-dataset perturbations, and controlled simulations. Figure 1 illustrates the joint estimation of question difficulty and model competency.

**Contributions.** This work makes three key contributions: 1) **A new evaluation framework.** We introduce *Empirical Interaction Propagation (EIP)*, a difficulty-aware and non-parametric approach that jointly estimates question difficulty and model competency. This allows evaluation to move beyond flat accuracy and provide finer-grained, difficulty-sensitive comparisons of LLM performance. 2) **Comprehensive empirical validation.** Through experiments on six benchmarks covering 35,550 questions and 30 models, EIP demonstrates strong alignment with human difficulty judgments (90% agreement), scalability to large evaluation settings, and robustness to changes in the model pool and dataset composition. 3) **New insights into models and datasets.** By applying EIP at scale, we reveal distinctive dataset difficulty profiles, consistent family-level patterns across model scales, and the reliability of open-weight and mixed-scale pools for producing stable difficulty estimates, offering practical guidance for benchmark design and model selection.

## 2 EMPIRICAL INTERACTION PROPAGATION (EIP)

**Method overview.** Question difficulty plays a central role in evaluating model performance. However, difficulty is inherently abstract and cannot be precisely defined. To address this, we operationalize difficulty through model failure: a question is considered difficult if even competent models are unable to solve it. EIP then evaluates question difficulty and measures model competency through an interactive process. EIP formulates questions and models as interdependent nodes within a directed bipartite graph, where questions' difficulty scores propagate to models that successfully solve questions, and models' competency score contributes to questions that they fail to answer correctly. We model this interconnection as an ergodic Markov chain defined on the bipartite graph.

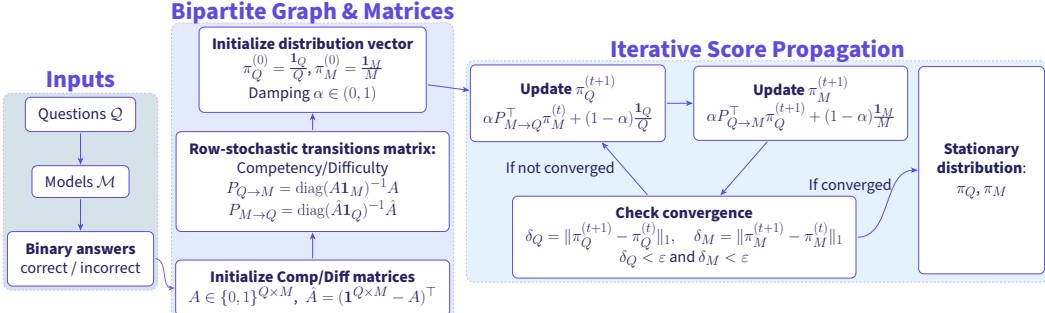

Figure 2: Detailed process demonstration of score propagation in EIP, which includes Evaluation Inputs, Bipartite Graph & Matrices, and Iterative Score Propagation.

We prove that this iterative process converges asymptotically to a unique stationary distribution, yielding a set of scores: $\pi_m$ for model competency and $\pi_q$ for question difficulty. A higher $\pi_m$ value signifies a model that consistently solves challenging questions, while a higher $\pi_q$ value indicates a question remains difficult even for competent models. These scores are mutually dependent, providing a self-reinforcing assessment of both models and questions.

## 2.1 SCORING IN EIP

We define the difficulty score $\pi_q$ for questions and the competency score $\pi_m$ for models based on the stationary distribution of a coupled random walk process. This formulation reflects the intuition that question difficulty is not an inherent property but rather emerges from the opposing patterns of model successes and failures. To enable bidirectional propagation between models and questions, we establish two reciprocal information flows. First, when model $m$ successfully answers question $q$, the difficulty weight of $q$ is distributed to $m$ in proportion to $q$'s total solvers. Conversely, when model $m$ fails to answer question $q$, its competency weight is transferred to $q$ in proportion to the total number of failures $m$ has encountered. Thus, the model competency $\pi_m$ and question difficulty $\pi_q$ are defined as:

$$\pi_m \propto \sum_{q \in \text{Success}(m)} \frac{\pi_q}{S(q)} \qquad \pi_q \propto \sum_{m \in \text{Fail}(q)} \frac{\pi_m}{F(m)} \tag{1}$$

where $S(q)$ represents the number of models that correctly solved $q$, and $F(m)$ denotes the total number of questions that model $m$ has failed.

## 2.2 DIRECTED BIPARTITE GRAPH AND PERFORMANCE MATRICES

We formalize LLM-question interactions using a directed bipartite graph $\mathcal{G} = (\mathcal{V}, \mathcal{E})$. The vertex set $\mathcal{V} = \mathcal{M} \cup \mathcal{Q}$ comprises $M$ models $\mathcal{M} = \{m_1, \ldots, m_M\}$ and $Q$ questions $\mathcal{Q} = \{q_1, \ldots, q_Q\}$ (typically $M \ll Q$). The edge set $\mathcal{E} = \mathcal{E}_{\text{Comp}} \cup \mathcal{E}_{\text{Fail}}$ signifies model performance: **Competency Edges** ($\mathcal{E}_{\text{Comp}}$): A directed edge $(q_i \to m_j) \in \mathcal{E}_{\text{Comp}} \subseteq \mathcal{Q} \times \mathcal{M}$ indicates that model $m_j$ correctly answered question $q_i$. **Difficulty Edges** ($\mathcal{E}_{\text{Fail}}$): Conversely, a directed edge $(m_j \to q_i) \in \mathcal{E}_{\text{Fail}} \subseteq \mathcal{M} \times \mathcal{Q}$ indicates that model $m_j$ failed to answer question $q_i$.

These interactions are encoded into two complementary binary matrices, constructed after filtering questions: **Competency Matrix** ($A$): This matrix $A \in \{0,1\}^{Q \times M}$ has entries $A_{ij} = \mathbb{I}[(q_i \to m_j) \in \mathcal{E}_{\text{Comp}}]$, where $A_{ij} = 1$ if model $m_j$ correctly answered question $q_i$. **Difficulty Matrix** ($\hat{A}$): This matrix $\hat{A} \in \{0,1\}^{M \times Q}$, expressed as $\hat{A} = (\mathbf{1}^{Q \times M} - A)^\top$, is the transposed complement of $A$. Its entries $\hat{A}_{ji} = \mathbb{I}[(m_j \to q_i) \in \mathcal{E}_{\text{Fail}}]$ indicate if model $m_j$ failed question $q_i$.

Prior to matrix construction, we exclude questions universally solved by all models or failed by all models, ensuring $0 < \sum_{j=1}^{M} A_{ij} < M$ for all retained questions. This step guarantees graph connectivity and avoids trivial solutions. Such extreme cases are rare (e.g., in our experiments with 35,550 questions and 30 models, only 2% of questions were universally solved or failed, while no models showed 100%/0% accuracy) and are assigned conceptual lowest/highest difficulty. These matrices map the graph's edges to linear operators.

### 2.3 Propagating Scores via Iterative Refinement

Having established the directed bipartite graph, we formalize the score propagation. EIP employs an iterative refinement process to mutually determine question difficulty scores ($\pi_Q$) and model competency scores ($\pi_M$). The core of this process relies on two row-stochastic transition matrices:

**Competency transition** ($P_{Q \to M}$): $P_{Q \to M} = \text{diag}(A\mathbf{1}_M)^{-1}A$. Given a question $q$, the probability of transitioning to a model $m$ is proportional to $m$'s success on $q$. The $q$-th row is normalized by $S(q)$, the number of models that correctly solved $q$.

**Difficulty transition** ($P_{M \to Q}$): $P_{M \to Q} = \text{diag}(\hat{A}\mathbf{1}_Q)^{-1}\hat{A}$. Given a model $m$, the probability of transitioning to a question $q$ is proportional to $m$'s failure on $q$. The $m$-th row is normalized by $F(m)$, the number of questions $m$ failed.

Scores are updated iteratively. An underlying random walk on the bipartite graph ($\mathcal{Q} \leftrightarrow \mathcal{M}$) would be 2-periodic. To ensure convergence to a unique stationary distribution, we introduce a damping factor $\alpha \in (0, 1)$, representing a "teleportation" probability. The iterative update equations for the scores at step $t + 1$ are:

$$\pi_Q^{(t+1)} = \alpha P_{M \to Q}^\top \pi_M^{(t)} + (1-\alpha)\frac{\mathbf{1}_Q}{Q} \quad (2) \qquad \pi_M^{(t+1)} = \alpha P_{Q \to M}^\top \pi_Q^{(t+1)} + (1-\alpha)\frac{\mathbf{1}_M}{M} \quad (3)$$

Here, $\mathbf{1}_Q$ and $\mathbf{1}_M$ are all-ones vectors of appropriate dimensions, $Q = |\mathcal{Q}|$, and $M = |\mathcal{M}|$. The transposes $P_{M \to Q}^\top$ and $P_{Q \to M}^\top$ facilitate score propagation. $\pi_M^{(t+1)}$ uses the just-updated $\pi_Q^{(t+1)}$.

This iterative process, where competency and difficulty scores are mutually reinforced, is guaranteed to converge to unique stationary distributions $\pi_Q$ and $\pi_M$. This is because the damped system forms an ergodic Markov chain, whose properties are established by the Perron-Frobenius theorem (Perron, 1907). A formal proof of existence, uniqueness, and convergence is provided in Appendix C.

### 2.4 Extending EIP to Continuous Scores

Benchmarks that grade free-form answers often provide partial credit rather than binary outcomes. EIP accommodates this setting by allowing the response matrix to take values in the unit interval. Let $A_c \in [0, 1]^{Q \times M}$ denote this continuous analogue of the correct-response matrix ($c$ stands for "**c**continuous"). All subsequent quantities mirror the binary case after replacing $A$ with $A_c$: The row sums $S(q) = \sum_{m=1}^{M}(A_c)_{qm}$ represent the total score achieved on question $q$, with questions satisfying $S(q) = 0$ removed as before. The complement matrix $\hat{A}_c = \mathbf{1}^{Q \times M} - A_c$ and corresponding column sums $F(m) = \sum_{q=1}^{Q}(\hat{A}_c)_{mq}$ maintain the same interpretation, with $F(m) > 0$ ensured in practice. The transition matrices retain the same form:

$$P_{Q \to M} = \text{diag}(S)^{-1}A_c \qquad (4) \qquad\qquad P_{M \to Q} = \text{diag}(F)^{-1}\hat{A}_c \qquad (5)$$

The iterative updates in Equation 2–3 and convergence guarantee of Theorem C.1 apply *unchanged*.

## 3 Experiment

### 3.1 Experiment Setup

**Datasets & benchmarks & models.** We select a diverse and representative set of datasets that encompasses diverse domains, including math, science, natural language understanding, and programming. We show the details of selected datasets in Table 1. We aggregate a diverse selection of 30 LLMs, covering a wide range of sizes from 0.5B to hundreds of billions parameters[1]. The models cover a diverse range and incorporate both open-source and proprietary models to ensure a comprehensive evaluation. The details of selected models are shown in Appendix H.

Table 1: Details of selected datasets.

| Dataset Name | # Questions |
|---|---|
| BBH (Suzgun et al., 2022) | 6511 |
| GPQA (Rein et al., 2023) | 646 |
| GSM8k (Cobbe et al., 2021) | 1319 |
| HellaSwag (Zellers et al., 2019) | 10042 |
| MATH (Hendrycks et al., 2021b) | 5000 |
| MMLU-Pro (Wang et al., 2024) | 12032 |

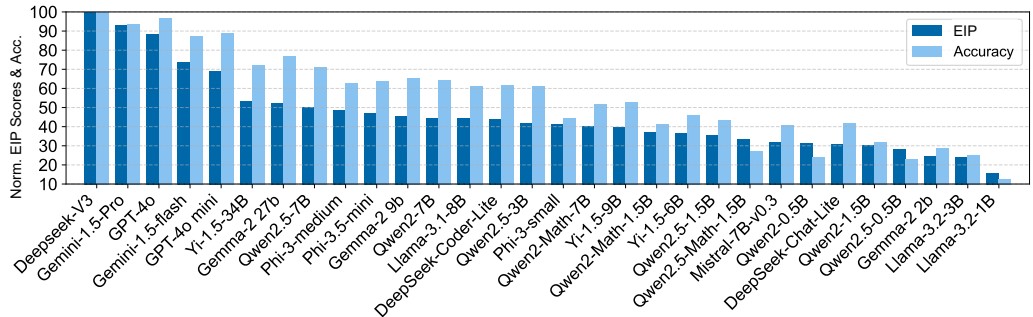

Figure 3: EIP scores and accuracies of models, both normalized to a fixed maximum of 100 to facilitate direct comparison. The full names of the models are provided in Appendix H.

## 3.2 MAIN RESULTS

**EIP offers finer-grained, difficulty-aware insights beyond traditional accuracy-based ranking.**
A key distinction between EIP and traditional accuracy-based evaluations lies in how question difficulty is incorporated. Traditional accuracy metrics treat all questions equally, which can lead to underestimating models whose overall accuracy is limited but whose strength lies in solving difficult questions. In contrast, EIP explicitly incorporates quantified question difficulty, enabling a more nuanced evaluation of model performance.

The positive correlation between EIP scores and accuracy (Kendall's Tau $\tau = 0.8492$; see Appendix G) aligns with our expectations, as stronger models typically answer more questions correctly and consequently receive higher scores. This validates that EIP serves as an enhancement to traditional accuracy-based ranking rather than a complete departure from it. However, Figure 3 reveals that significant discrepancies in model rankings frequently emerge, particularly among adjacent models. While the overall trend preserves the general hierarchy between clearly superior and inferior models, substantial rank changes frequently occur between closely-performing models, highlighting EIP's ability to provide finer-grained differentiation. In particular, models with high raw accuracy may receive lower EIP scores than their neighbors, while some lower-accuracy models are ranked higher—reflecting their stronger performance on more difficult questions. For instance, Qwen2-0.5B (20.2%) is ranked higher than DeepSeek-Chat-Lite (30.49%). This re-ranking stems from EIP's central design: assigning greater credit to correct answers on more challenging questions. Figure 11 supports this observation: Qwen2-0.5B correctly answered 5.5% of hard questions, compared to only 2.4% for DeepSeek-Chat-Lite, which substantially boosts its EIP score. A similar pattern is observed among top-performing models: while GPT-4o achieves higher overall accuracy, Gemini-1.5-Pro receives a higher EIP score due to answering 5% more medium- and high-difficulty questions. This illustrates how EIP identifies strengths on difficult questions that are otherwise obscured by flat accuracy scores—a finding consistent with our simulated Case Study (section 4).

**Model families maintain consistent difficulty patterns despite scaling effects.** As shown in Figure 5, our leave-one-out[2] evaluation demonstrates that models within the Llama-3 family (Llama-3.1-8B, Llama-3.2-3B-Instruct, Llama-3.2-1B-Instruct) exhibit stable difficulty-specific answering distributions across different parameter scales. Despite substantial differences in overall accuracy ($51.2\% \rightarrow 21.0\% \rightarrow 10.2\%$) and EIP scores ($44.3\% \rightarrow 23.9\% \rightarrow 15.4\%$), all variants preserve a similar distribution of correct responses across difficulty levels (hard / medium / easy). Notably, the 8B model demonstrates a slight disadvantage in hard-question accuracy (11.1% compared to 15.7% and 15.6% for the smaller variants), yet maintains nearly identical easy-question accuracy (62.9% compared to 60.4% and 61.5% for the smaller variants). This trend extends across other model families, including Qwen and Yi (Appendix F), indicating that: 1) Accuracy scaling primarily influences absolute performance rather than altering relative difficulty distributions. 2) Traditional accuracy metrics obscure essential behavioral consistencies across model scales.

---

[1]In EIP, a model is categorized as small if it has fewer than 10B parameters, medium if its size ranges between 10B and 50B parameters, and large if it exceeds 50B parameters.

[2]Leave-One-Out is a cross-validation method that iteratively uses each sample in the dataset as the validation set, while the remaining samples form the training set.

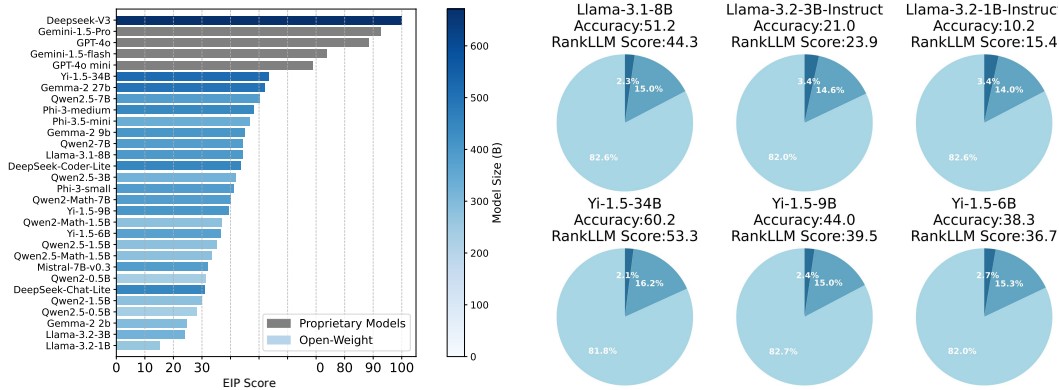

Figure 4: Relationship between model performance and parameter scale.

Figure 5: The proportion of Easy/Medium/Hard questions within correctly answered samples across Llama/Yi variants.

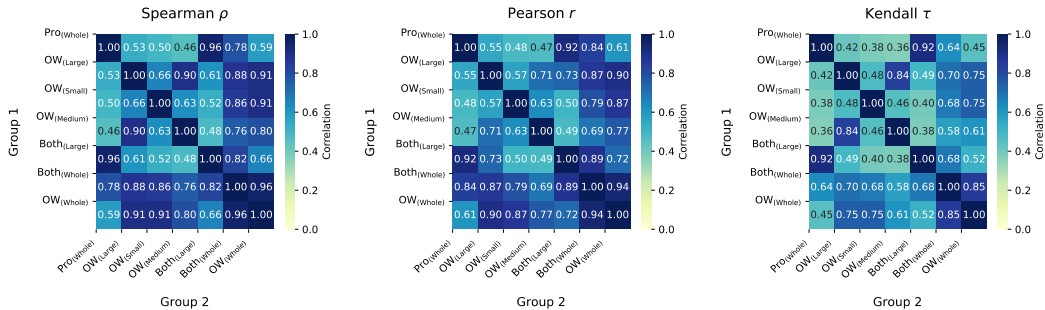

Figure 6: Correlation analysis across model group combinations (Pro. means proprietary, and OW means open weight).

**Open-weight models show strong potential in estimating difficulty, rivaling proprietary models.** To evaluate the consistency of question difficulty distributions across model groups, we employed three correlation methods: Spearman, Pearson, and Kendall. These methods, capturing monotonic, linear, and rank-based correlations respectively, yielded highly consistent results, indicating strong agreement in the difficulty rankings produced by different model sets. As shown in Figure 6, the difficulty distributions derived from various model groups exhibit significant correlations. The difficulty distributions generated by open-weight models of all sizes ($OW_{(Whole)}$) demonstrate a strong positive correlation with those generated by the entire set of models ($Both_{(Whole)}$). Quantitatively, the Spearman, Pearson, and Kendall correlations are 0.96, 0.94, and 0.85, respectively. These high correlation values across all three metrics indicate a strong agreement between the difficulty rankings produced by open-weight models and those produced by the full model set, suggesting that open-weight models alone are capable of capturing the overall difficulty landscape as represented by EIP. The difficulty distributions derived from all proprietary models ($Pro_{(Whole)}$) show a correlation with those derived from all models ($Both_{(Whole)}$), with a correlation of 0.78, 0.84, and 0.64. While still demonstrating agreement, there is a noticeable drop in the Spearman correlation compared to the open-weight models.

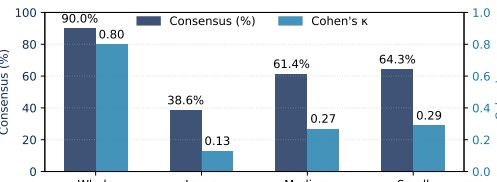

Figure 7: Consensus alignment with human judgments on EIP-estimated question difficulty by model size group.

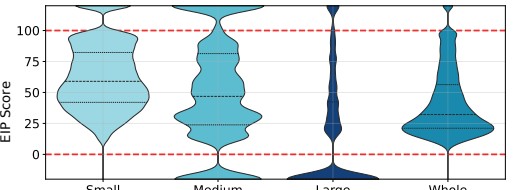

Figure 8: Difficulty distribution grouped by model size.

Table 2: Alignment analysis with human evaluation on question difficulty (all metrics in %).

| Method | $V_1$ | $V_2$ | $V_3$ | $V_4$ | $V_5$ | $V_6$ | $V_7$ | $V_8$ | $V_9$ | $\cdots$ | $V_{20}$ | Consensus (%) |
|---|---|---|---|---|---|---|---|---|---|---|---|---|
| Simple Rank | 41.4 | 54.3 | 50.0 | 60.0 | 51.4 | 54.3 | 54.3 | 50.0 | 50.0 | $\cdots$ | 52.9 | 62.9 |
| 1PL-IRT | 37.1 | 44.3 | 40.0 | 52.9 | 51.4 | 44.3 | 45.7 | 45.7 | 48.6 | $\cdots$ | 41.4 | 50.0 |
| 2PL-IRT | 35.7 | 45.7 | 41.4 | 54.3 | 52.9 | 42.9 | 44.3 | 47.1 | 47.1 | $\cdots$ | 42.9 | 51.4 |
| Multi-IRT | **50.0** | 50.0 | 48.6 | 54.3 | 48.6 | 57.1 | 47.1 | 47.1 | 44.3 | $\cdots$ | 48.6 | 52.9 |
| **EIP** | 37.1 | **70.0** | **62.9** | **71.4** | **67.1** | **61.4** | **74.3** | **64.3** | **57.1** | $\cdots$ | **62.9** | **90.0** |

**A diverse model pool mitigates extreme question-difficulty estimates in EIP.** As shown in Figure 8[3], we analyze the impact of model diversity on difficulty estimation by running EIP with four different annotator model pools: small-only, medium-only, large-only, and a combination of all (Whole). Our findings highlight that the composition of the model pool used by EIP plays a crucial role in determining the quality of difficulty estimation. Homogeneous model groups tend to exhibit more extreme difficulty distributions. Specifically, the Large group classifies 58% of questions as "overly easy", while the Small and Medium groups contain over 30% "impossible" questions. These extremes introduce a high proportion of "dead nodes" in EIP's algorithm, where models either consistently overestimate or underestimate question difficulty. In contrast, the Whole group, which includes models of varying scales, achieves a more balanced difficulty distribution. Complementary error patterns across different model scales reduce extreme cases to below 3%. Moreover, the mixed-scale ensemble maintains meaningful bidirectional score propagation in EIP, reducing dead nodes by 83% compared to homogeneous groups. These results align with the "wisdom of the crowd" principle, demonstrating that incorporating diverse models not only mitigates mutual biases but also preserves sufficient granularity to prevent systematic overestimation or underestimation in difficulty assessment.

**Difficulty distribution varies by dataset.** As shown in Figure 9, the six benchmarks exhibit distinct difficulty profiles. GPQA has a relatively uniform distribution, making it ideal for evaluating performance across a range of difficulty levels. In contrast, GSM8k, HellaSwag, and BBH are skewed toward easier questions, with most items concentrated at lower EIP difficulty scores. This skew may limit their utility in assessing performance on harder tasks but provides insights into models' fundamental capabilities. MATH and MMLU-Pro, on the other hand,

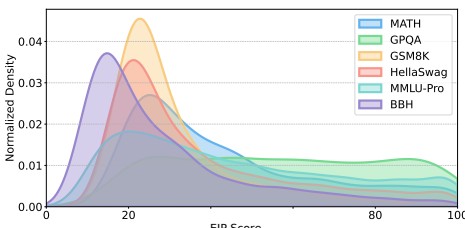

Figure 9: Question difficulty distributions for six benchmarks.

display broader and more dispersed distributions, with MATH emphasizing harder questions and MMLU-Pro displaying a bimodal pattern, blending challenging items. These benchmarks, therefore, are better suited for evaluating advanced reasoning capabilities and the robustness of models.

### 3.3 HUMAN EVALUATION

To validate the alignment of EIP with human judgment, we conducted a controlled blind trial involving 20 human evaluators ($V_1$–$V_{20}$). Each evaluator compared two questions from the same subject category (e.g., mathematics) to judge which was more difficult. There are 70 randomly ordered question pairs in total. A detailed evaluation protocol is in subsection H.3.

**Baseline**. As comparisons, we selected the Simple Rank method as a primary baseline. Additionally, we evaluated three Item Response Theory (IRT) based models: 1pl-irt, 2pl-irt, and multi-irt. Configuration details for these IRT models are provided in subsection H.4. Simple Rank determines question difficulty based solely on the number of models that answered incorrectly, assigning higher difficulty to questions with higher error counts.

**EIP achieves high alignment (90%) with human consensus.** As shown in Table 2, EIP achieves superior agreement with human judgments for 18 out of 20 evaluators compared to Simple Rank, with an average margin of +9.3% in individual alignment scores. The "Consensus" column in Table 2 aggregates information via majority vote and resolves near-ties. Overall, EIP achieves 90% agreement with the human consensus, highlighting its strong alignment with human judgment.

**A more diverse model pool improves human alignment in difficulty judgments.** As shown in Figure 7, the whole group, comprising models with various sizes, achieves a consensus alignment of

---

[3]Area above 100 represents questions that all models failed, below 0 are all correct answers.

90.0% with human judgments on question difficulty, along with a Cohen coefficient of $\kappa$ of 0.80. In contrast, the homogeneous size groups from the SOTA large model to the 0.5B small models exhibit markedly lower consensus (38.6%–64.3%) and weaker agreement levels ($\kappa = 0.13$–0.29). This strongly suggests that diversity in model size leads to richer and more robust difficulty assessments, reducing the bias often observed when the annotator pool is restricted to a single-size scale.

### 3.4 Efficiency and Scalability Analysis of EIP

Given the rapidly growing number of models ($M$) and extensive question sets ($Q$) used in benchmarks, computational efficiency and scalability of the evaluation framework are paramount. We therefore analyze these aspects of EIP both theoretically and empirically.

**EIP demonstrates strong computational efficiency and scalability.** EIP has time complexity $O(tQM)$ where $t$ is the number of iterations required for convergence (proof in Appendix D). The iteration count $t$ remains stable with preset damping factor $\alpha$ and tolerance $\epsilon$, independent of $Q$ or $M$ (Appendix C). Since $M \ll Q$ in typical LLM evaluation scenarios, the per-iteration cost $O(QM)$ remains manageable for large-scale applications.

Empirical validations further underscore EIP's efficiency and scalability. In a direct comparison on a dataset of 30 LLMs and 35,550 questions, EIP achieved convergence substantially faster (in 0.00597 seconds) than all Item Response Theory (IRT) baselines, including 1PL-IRT, 2PL-IRT, and Multi-IRT (Table 3). Scalability was further assessed using synthetic response matrices with $Q$ ranging up to 1,000,000 and $M$ up to 2,000. Across all these set-

Table 3: Convergence speed of EIP on consumer-grade hardware (Intel i7).

| Method | Convergence Time (s) |
|---|---|
| EIP | **0.00597** |
| 1PL IRT | 1782.75 |
| 2PL IRT | 3787.03 |
| Multi-IRT (3D) | 18.76 |

tings, EIP consistently converged in a constant number of iterations (9 in our experiments), and its average per-iteration time increased linearly with the total number of interactions $Q \times M$, aligning with its theoretical complexity (Table 4).

These combined theoretical and empirical results confirm that EIP not only converges quickly but also scales efficiently with the size of the evaluation. Given the low computational overhead, deployment costs are negligible even for extreme-scale evaluations. On AWS EC2 (approximately \$0.05 per vCPU-hour), EIP enables **cost-efficient deployment for large-scale LLM evaluation**. We have established a Hugging Face leaderboard platform that supports continuous updates of new models and benchmarks, capable of handling hundreds of daily updates while maintaining cost efficiency. This makes EIP a practical and robust solution for ranking large numbers of language models across extensive question sets.

Table 4: EIP convergence time under varying numbers of models and questions.

| Questions ($Q$) | Models ($M$) | $Q \times M$ | Iterations ($T$) | Total Time (s) | Avg. Time / Iter. (s) |
|---|---|---|---|---|---|
| 1,000,000 | 2,000 | $2.00 \times 10^9$ | 9 | 5.28 | 0.5867 |
| 1,000,000 | 1,000 | $1.00 \times 10^9$ | 9 | 3.14 | 0.3489 |
| 500,000 | 1,000 | $0.50 \times 10^9$ | 9 | 1.93 | 0.2144 |
| 1,000,000 | 500 | $0.50 \times 10^9$ | 9 | 2.18 | 0.2422 |
| 500,000 | 500 | $0.25 \times 10^9$ | 9 | 1.13 | 0.1256 |
| 500,000 | 250 | $0.125 \times 10^9$ | 9 | 0.64 | 0.0711 |
| 250,000 | 500 | $0.125 \times 10^9$ | 9 | 0.52 | 0.0578 |
| 250,000 | 250 | $0.0625 \times 10^9$ | 9 | 0.30 | 0.0333 |

### 3.5 Robustness and Extensibility Analysis of EIP

Given the rapid pace at which new models are developed and integrated into evaluation pipelines, it is crucial that evaluation systems remain both stable and extensible. To this end, we examine the robustness and extensibility of EIP under dynamic model pool configurations, ensuring that its scores remain consistent even as models are added. To simulate the introduction of new models, we emulate pool variation by randomly removing $k$ models (ranging from 1 to 15) from the original set of 30 models. For each value of $k$, we conducted 50 independent trials and computed the mean and standard deviation of the Spearman correlation $\rho$ between the resulting scores and those obtained with the full model pool.

**Even under large-scale modifications to the model pool, EIP preserves stable rankings of both questions and models.** This is substantiated by the results in Table 5 and visualized in Figure 10. Even when half of the model pool (15 out of 30 models) was removed—simulating a scenario where a large influx of new models suddenly enters the system—the mean $\rho$ for question difficulty scores remained high at $0.9382$ when compared to rankings from the full pool. Model competency rankings exhibited even greater resilience under the same 15-model removal condition. These findings indicate that EIP's relative assessments of both questions and models are largely preserved despite considerable variations in the evaluation pool's size. Concurrently, a significant practical benefit observed was the reduction in computation time, demonstrating improved efficiency with smaller model sets without a critical loss in ranking fidelity. For other detailed experiments and observations such as the removal of datasets, please refer to Appendix E.

Table 5: Impact of randomly removing $k$ models on EIP's stability and computation time (averaged over 50 trials).

| Models Removed ($k$) | Question Correlation | | Model Correlation | | Computation Time | |
|---|---|---|---|---|---|---|
| | Mean $\rho$ | SD | Mean $\rho$ | SD | Avg. Time (s) | % Reduction |
| 1 | 0.9978 | 0.0021 | 0.9997 | 0.0002 | 0.0079 | 14.09 |
| 5 | 0.9850 | 0.0073 | 0.9988 | 0.0009 | 0.0080 | 13.57 |
| 10 | 0.9684 | 0.0107 | 0.9977 | 0.0019 | 0.0066 | 28.28 |
| 15 | 0.9382 | 0.0187 | 0.9934 | 0.0069 | 0.0062 | 33.02 |

## 4 CASE STUDY: VALIDATING DIFFICULTY-BASED EVALUATION

Due to the lack of ground truth in large-scale evaluations, identifying model capabilities on difficult questions is challenging. We address this through a controlled simulation where model performance is defined by question difficulty, enabling clear assessment of how different evaluation methods capture performance differences across difficulty levels.

**Simulation Setup.** We created five hypothetical models (M1–M5) and 100 questions (70 easy, 21 medium, 9 hard) to assess whether evaluation methods can detect subtle performance differences among models with similar overall accuracy. Models were configured to establish ground truth ranking: M1 > M2 > M4 > M5 > M3. M1/2 had equal accuracy, but M1 answered more hard questions. M4/5 had equal accuracy, but M4 performed better on medium-difficulty items.

**Baseline.** We applied EIP alongside baseline methods to simulated responses: **1) Standard accuracy:** Raw percentage of correct answers. **2) Dataset-difficulty-weighted accuracy:** Weighted average of accuracies on "Easy" and "Hard" subsets, with weight $w_d = 1 - \bar{a}_d$ for dataset $d$, where $\bar{a}_d$ is average accuracy of all models on that dataset. **3) Item Response Theory ((Van der Linden, 2018)):** 1PL, 2PL, and multidimensional IRT models. Configuration details are in subsection H.4.

Table 6: Model rankings under different evaluation methods, with ✓ indicating agreement with the ground truth ranking.

| Model | Solved Qs | EIP | | Accuracy | | WeightedScore | | 1PL-IRT | | 2PL-IRT | | Multi-IRT | | Truth |
|---|---|---|---|---|---|---|---|---|---|---|---|---|---|---|
| | | Score | Rank | Score | Rank | Score | Rank | Score | Rank | Score | Rank | Score | Rank | Rank |
| M1 | 70/10/5 | 100.00 | 1 | 85 | 1 | 77.01 | 2 | 100.00 | 1 | 100.00 | 1 | 38.91 | 3 | 1 |
| M2 | 70/11/4 | 95.50 | 2 | 85 | 1 | 78.79 | 1 | 100.00 | 1 | 84.14 | 2 | 0.00 | 5 | 2 |
| M3 | 47/13/0 | 56.88 | 5 | 60 | 5 | 58.86 | 3 | 0.00 | 5 | 19.66 | 3 | 95.30 | 2 | 5 |
| M4 | 46/15/0 | 59.85 | 3 | 61 | 3 | 50.28 | 5 | 3.02 | 3 | 0.00 | 5 | 100.00 | 1 | 3 |
| M5 | 47/14/0 | 58.21 | 4 | 61 | 3 | 58.86 | 3 | 3.02 | 3 | 19.30 | 4 | 16.52 | 4 | 4 |
| M1>M2 Correct? | | ✓ | | ✗ | | ✗ | | ✗ | | ✓ | | ✗ | | |
| M4>M5>M3 Correct? | | ✓ | | ✗ | | ✗ | | ✓ | | ✗ | | ✗ | | |

Only EIP achieved rankings consistent with ground truth (Table 6). EIP distinguished M1 from M2 by capturing hard question performance differences and ordered M4 > M5 > M3 based on medium-difficulty distinctions. Baseline methods showed limitations: **Standard accuracy** treats all questions equally, failing to distinguish models with identical accuracy but different hard question competency. **Dataset-weighted accuracy** uses coarse-grained weighting

Table 7: Difficulty scores of simulated questions calculated by EIP.

| Category | Quantity | Mean | $\sigma$ |
|---|---|---|---|
| Easy | 70 | 18.24 | 0.45 |
| Medium | 21 | 73.44 | 1.56 |
| Hard | 9 | 98.59 | 1.34 |

that ignores intra-dataset variation, allowing models to score high by answering easier items within "hard" sets. **IRT-based methods** (1PL, 2PL, Multi-IRT) showed inconsistent rankings, with Multi-IRT diverging significantly from ground truth.

**Analysis of Simulation Results.**

These findings highlight EIP's unique ability to capture fine-grained performance differences through instance-level difficulty estimation, while conventional methods fail due to uniform weighting assumptions, aggregation bias, or insufficient data robustness.

## 5    CONCLUSION

We introduced EIP, a difficulty-aware framework for evaluating large language models. By jointly modeling question difficulty and model competence, EIP provides more fine-grained insights than traditional accuracy-based metrics. It achieves 90% agreement with human difficulty judgments, surpasses IRT-based and heuristic baselines, and maintains stable rankings under model or dataset perturbations. Moreover, it converges efficiently in large-scale settings. These results establish EIP as a scalable, robust, and human-aligned alternative for LLM evaluation.

## 6    ACKNOWLEDGEMENT

We thank Hanchi Sun for mathematical support, including insights on the random-walk formulation and help with the derivations and proofs underlying our damped bidirectional score propagation mechanism. We thank Yuan Guo for early involvement in the project and for contributions to the initial exploration and prototyping that informed this work. We also thank Guanshen Meng for assistance with the visual design of our figures, including color palette refinement and aesthetic improvements. This work is also supported by the Delta/DeltaAI systems at NCSA and the Bridges-2 system at PSC through allocation CIS240308 from the Advanced Cyberinfrastructure Coordination Ecosystem: Services & Support (ACCESS) program, supported by National Science Foundation grants 2138259, 2138286, 2138307, 2137603, and 2138296.

REPRODUCIBILITY STATEMENT

We are committed to ensuring that our results can be independently verified and extended. All source code and prompt templates used in our experiments are included in the supplementary materials. The repository includes: (i) a step-by-step README describing the environment setup and execution commands; (ii) hosted links to all model version and formatted benchmark datasets; and (iii) configuration files specifying random seeds, hyper-parameters, and compute resources. Human-evaluation protocols are likewise included. Together, these artifacts enable an end-to-end replication of our experiments on any machine with standard GPU resources.

ETHICS STATEMENT

EIP is an evaluation framework and does not generate new user-facing content. All benchmarks employed are publicly available and distributed under permissive licenses. No personally identifiable or sensitive data are processed. Nonetheless, difficulty-aware rankings could conceivably be misused to dismiss models that prioritize safety or fairness over raw competence. We therefore release our code under an open license and encourage responsible adoption that complements, rather than replaces, broader alignment assessments. We disclose all model weights evaluated and provide guidelines for reproducing the study without circumventing dataset usage terms.

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

# A APPENDIX

## APPENDIX CONTENTS

## B  RELATED WORKS

**Benchmarking and evaluating LLMs.** Owing to the advanced capabilities of LLMs, sophisticated benchmarking is essential to comprehensively assess their performance in both general and specialized domains (Chang et al., 2024). The evaluation of LLMs spans multiple fields, beginning with traditional core natural language processing (NLP) tasks such as sentiment analysis (Lopez-Lira & Tang, 2023; Zhang et al., 2023), text classification (Yang & Menczer, 2024; Zhang et al., 2024), and natural language inference (McKenna et al., 2023). In recent years, the development of diverse benchmarks has significantly expanded aspects of evaluating LLMs, enabling more comprehensive assessments of their capabilities. For example, the MMLU (Hendrycks et al., 2021a) and MMLU-Pro (Wang et al., 2024) assess models on multi-disciplinary language understanding and reasoning, and MMLU-Reason further benchmarks multi-task multi-modal language understanding and reasoning (Tie et al., 2025), while Big-Bench Hard (BBH) (Suzgun et al., 2022) evaluates their multi-step reasoning and algorithmic reasoning abilities. The GSM8k (Cobbe et al., 2021) benchmark tests performance on grade-school-level mathematics, and HumanEval (Chen et al., 2021) measures a model's capability in algorithmic reasoning. AlignBench is designed to evaluate the alignment of Chinese LLMs (Liu et al., 2023a; 2024), and HellaSwag (Zellers et al., 2019) focuses on commonsense reasoning by requiring LLMs to select the most plausible continuation of a given context. Guo et al. (2023) and Huang et al. (2024d) assess the abilities of LLMs in science domains, and recent benchmarks also extend evaluation to medicine with contamination-aware protocols and automated rubrics (Yan et al., 2026) and to structure-based drug design with systematic trade-offs across representations (Zheng et al., 2026). With the development of agentic AI, there are many benchmarks designed to evaluate the agent-related capabilities of LLMs (Liu et al., 2023b; Chen et al., 2024c;b; Huang et al., 2023a; 2024a), and robust evaluation beyond memorization has also been studied for vision-language-action models (Zhou et al., 2025). Some benchmarks are also focusing on evaluating the trustworthiness of LLMs (Huang et al., 2024c; Wang et al., 2023; Gao et al., 2024; Liu et al., 2025; Huang et al., 2023b; 2024b), and benchmarking unlearning is another complementary direction for trustworthy evaluation (Ma et al., 2025). Besides capability evaluation, some recent works propose many evaluation paradigms, including multimodal judge-based evaluation (Chen et al., 2024a) and broader perspectives on AI scientist systems that motivate evaluating complex agentic workflows (Anonymous, 2025). For instance, flexible protocols for dynamic evaluation have been advanced, exemplified by the recent initiatives DyVal (Zhu et al., 2023) and DyVal 2 (Zhu et al., 2024). Wu et al. (2024) and Bao et al. (2024) both propose a dynamic evaluation framework powered by generative models. Beyond dynamic protocols, fluid and continuously updated benchmarks have been proposed to better track distribution shift and rapid model progress (Hofmann et al., 2025). Psychometric alternatives based on adaptive testing further aim to reduce evaluation cost while maintaining measurement fidelity (Li et al., 2025), and generative computerized adaptive testing extends this direction by generating or selecting items online (Feng & Lan, 2026). Recent IRT-based analyses revisit benchmark conclusions by explicitly modeling item parameters (Zhou et al., 2026). In preference-based evaluation, rankings can be highly sensitive to small perturbations of comparison sets (Huang et al., 2026), motivating more robust inference from preference data (Frauen et al., 2026) and reliability diagnosis of LLM-as-a-judge using IRT (Choi et al., 2026). Finally, task-specific benchmarks extend evaluation to structured settings such as optimization modeling with graph-theoretic scoring (Wang et al., 2026), graph-structured diagram generation (Liang & You, 2025), and open-source game environments (Sistla et al., 2025).

**Difficulty measuring.** Most previous works rely on human effort to classify question difficulty levels. For example, in benchmarks like MATH (Hendrycks et al., 2021b) and GPQA (Rein et al., 2023), human experts annotate questions into predefined difficulty tiers. LingOly (Bean et al., 2024), a linguistic reasoning benchmark, categorizes question difficulty into five levels by considering both semantic similarity to English and the complexity of required reasoning. The labeling process is time-consuming and lacks scalability. Unlike human labeling, OlympicArena (Huang et al., 2024e) employs GPT-4V as an annotator to categorize question difficulty levels. MedConceptsQA (Shoham & Rappoport, 2024), on the other hand, determines difficulty levels based on the relative distances among answer choices within an undirected graph built by medical code. However, these approaches either introduce bias in difficulty assessment due to reliance on a single model or are domain-specific, limiting their generalizability to other fields. Recent work also estimates question difficulty from model-internal signals, e.g., hidden representations capturing LLM-perceived difficulty (Zhu et al., 2025). Complementarily, difficulty can be inferred without ground truth by leveraging comparative

signals across LLM outputs (Ballon et al., 2025). Recent work, Easy2Hard-Bench (Ding et al., 2024), applied two independent methods to quantify question difficulty: Item Response Theory (IRT) (Lord & Novick, 2008) and Glicko-2 (Glickman, 2012). Unlike IRT, which assumes static difficulty parameters, EIP dynamically adjusts scores based on model performance, providing a more adaptive and accurate representation of difficulty. In contrast to Glicko-2, which relies on pairwise matchups, EIP employs a bidirectional score propagation approach, allowing question difficulty to be inferred from a broader set of solver interactions rather than isolated 'matches', which provides a dataset-wide estimate of difficulty.

## C  PROOF OF CONVERGENCE

**Convergence**. Let $\Delta^{(t)} = \|\pi_Q^{(t)} - \pi_Q^{(t-1)}\|_1 + \|\pi_M^{(t)} - \pi_M^{(t-1)}\|_1$ measure the change between iterations. The algorithm terminates when $\Delta^{(t)} < \epsilon$ for predefined tolerance $\epsilon$. For irreducible chains, convergence occurs at a geometric rate $O(\alpha^t)$, typically requiring $\log\frac{1}{\epsilon}$ iterations. An important consideration is the convergence speed in large-scale settings. In our experiments with 30 models and 35,550 questions, we find that about 9 iterations of the bipartite propagation process are sufficient to reach a stable distribution of difficulty and competency scores. Each iteration primarily involves matrix-vector multiplications on $A$ and $\hat{A}$, making the approach computationally feasible even with tens of thousands of questions. The relation between the damping factor $\alpha$ and the speed of convergence is shown in Table 8.

Table 8: $\Delta$ for different damping factors ($\alpha$) across iterations, indicating the difference between the stationary distribution and convergence.

| Iteration | $\alpha$=0.10 | $\alpha$=0.20 | $\alpha$=0.30 | $\alpha$=0.40 | $\alpha$=0.50 | $\alpha$=0.60 | $\alpha$=0.70 | $\alpha$=0.80 | $\alpha$=0.90 | $\alpha$=1 |
|---|---|---|---|---|---|---|---|---|---|---|
| 0 | 8.39e-02 | 1.68e-01 | 2.52e-01 | 3.37e-01 | 4.22e-01 | 5.07e-01 | 5.93e-01 | 6.80e-01 | 7.67e-01 | 8.55e-01 |
| 1 | 1.86e-03 | 7.65e-03 | 1.77e-02 | 3.24e-02 | 5.20e-02 | 7.70e-02 | 1.08e-01 | 1.44e-01 | 1.88e-01 | 2.38e-01 |
| 2 | 1.20e-06 | 2.00e-05 | 1.06e-04 | 3.48e-04 | 8.84e-04 | 1.91e-03 | 3.66e-03 | 6.49e-03 | 1.08e-02 | 1.70e-02 |
| 3 | 6.11e-10 | 4.14e-08 | 5.00e-07 | 2.97e-06 | 1.20e-05 | 3.76e-05 | 9.99e-05 | 2.34e-04 | 4.99e-04 | 9.84e-04 |
| 4 | 4.59e-13 | 1.19e-10 | 3.10e-09 | 3.17e-08 | 1.94e-07 | 8.54e-07 | 3.01e-06 | 9.02e-06 | 2.38e-05 | 5.70e-05 |
| 5 | Converge | 6.28e-13 | 3.69e-11 | 6.63e-10 | 6.26e-09 | 3.92e-08 | 1.86e-07 | 7.16e-07 | 2.36e-06 | 6.85e-06 |
| 6 | Converge | Converge | 3.41e-13 | 1.10e-11 | 1.65e-10 | 1.50e-09 | 9.78e-09 | 4.97e-08 | 2.09e-07 | 7.58e-07 |
| 7 | Converge | Converge | Converge | 1.44e-13 | 3.33e-12 | 4.42e-11 | 3.94e-10 | 2.64e-09 | 1.41e-08 | 6.37e-08 |
| 8 | Converge | Converge | Converge | Converge | 5.38e-14 | 1.03e-12 | 1.27e-11 | 1.12e-10 | 7.67e-10 | 4.30e-09 |
| 9 | Converge | Converge | Converge | Converge | Converge | Converge | 3.27e-13 | 3.79e-12 | 3.32e-11 | 2.32e-10 |
| 10 | Converge | Converge | Converge | Converge | Converge | Converge | Converge | 1.74e-13 | 1.87e-12 | 1.58e-11 |

**Theorem C.1** (Existence and Uniqueness of the Stationary Distribution). *Let $P_{Q\to M} \in \mathbb{R}^{Q\times M}$ and $P_{M\to Q} \in \mathbb{R}^{M\times Q}$ be the row-stochastic matrices defined in subsection 2.3. For any damping factor $\alpha \in (0,1)$, the iterative process defined in Equation 2 and Equation 3 converges to a unique stationary distribution $(\pi_Q, \pi_M)$ satisfying*

$$\pi_M = \alpha\, P_{Q\to M}^\top \pi_Q + (1-\alpha)\,\frac{\mathbf{1}_M}{M}, \tag{6}$$

$$\pi_Q = \alpha\, P_{M\to Q}^\top \pi_M + (1-\alpha)\,\frac{\mathbf{1}_Q}{Q}. \tag{7}$$

*Furthermore, $\pi_Q$ and $\pi_M$ have strictly positive entries.*

*Proof sketch of Theorem C.1.* **Step 1: Reformulation as a Single Markov Chain.** The alternating updates between $\pi_Q$ and $\pi_M$ in Equation 6 and Equation 7 can be unified into a single Markov chain on an augmented state space of size $Q + M$. Define the combined state vector $\pi^{(t)} = \begin{bmatrix} \pi_Q^{(t)} \\ \pi_M^{(t)} \end{bmatrix}$. The iterative updates become: $\quad \pi^{(t+1)} = \alpha\mathcal{P}^\top \pi^{(t)} + (1-\alpha)\mathbf{v},$ where the block matrix $\mathcal{P} = \begin{bmatrix} 0 & P_{Q\to M} \\ P_{M\to Q} & 0 \end{bmatrix}$ preserves the bipartite transitions, and $\mathbf{v} = \begin{bmatrix} \frac{\mathbf{1}_Q}{Q} \\ \frac{\mathbf{1}_M}{M} \end{bmatrix}$ represents uniform teleportation.

**Step 2: Damped Markov chain interpretation.** The combined dynamics correspond to a Markov chain with a transition matrix:

$$T = \alpha \mathcal{P}^\top + (1 - \alpha)\mathbf{v}\mathbf{1}_{Q+M}^\top,$$

where $\mathbf{1}_{Q+M}$ is the all-ones vector. At each step, the chain either follows $\mathcal{P}^\top$ (with probability $\alpha$) or restarts uniformly via $\mathbf{v}$ (with probability $1 - \alpha$), mirroring the damping mechanism in Equation 6–Equation 7.

**Step 3: Irreducibility and aperiodicity.** 1) *Irreducibility*: The teleportation term ensures every state can reach any other state in one step, as $(1 - \alpha)\mathbf{v} > 0$ componentwise. 2) *Aperiodicity*: Self-transitions occur with probability at least $(1 - \alpha)\min\left(\frac{1}{Q}, \frac{1}{M}\right) > 0$, breaking periodicity inherent in the bipartite structure. Thus, $T$ is irreducible and aperiodic.

**Step 4: Perron-Frobenius theorem application.** By the Perron-Frobenius theorem for irreducible and aperiodic Markov chains, $T$ has a unique stationary distribution $\pi = \begin{bmatrix} \pi_Q \\ \pi_M \end{bmatrix}$ with $\pi > 0$, satisfying $\pi = T^\top \pi$. Substituting $T$ into this equation recovers the coupled updates in Equation 6–Equation 7, confirming $(\pi_Q, \pi_M)$ as the unique solution. Crucially, convergence iteration $t$ depends on the damping factor $\alpha$ and the desired precision $\epsilon$, but it is **independent of the scale of the problem, i.e., the number of questions** $Q$ **or models** $M$. $\qquad\square$

This proof rigorously establishes Theorem C.1 using notation consistent with the main text. The use of $\mathcal{P}$, $\mathbf{v}$, and damping factor $\alpha$ directly corresponds to the iterative updates in Equation 6–Equation 7, ensuring notational coherence.

# D    TIME COMPLEXITY ANALYSIS

In this section, we calculate and prove the time complexity of EIP in theory.

Let $Q$ be the number of questions and $M$ be the number of models. The Competency Matrix is denoted by $A \in \{0, 1\}^{Q \times M}$, and the Difficulty Matrix (transposed) is $\hat{A} = (\mathbf{1}^{Q \times M} - A)^\top \in \{0, 1\}^{M \times Q}$. The transition matrix $P_{Q \to M} \in \mathbb{R}^{Q \times M}$ is derived from $A$, and $P_{M \to Q} \in \mathbb{R}^{M \times Q}$ is derived from $\hat{A}$.

Each iteration of EIP primarily involves two sparse matrix-vector multiplications as defined in Equation 2 and Equation 3 of the main paper. The operation $P_{M \to Q}^\top \pi_M^{(t)}$ involves multiplying the transpose of $P_{M \to Q}$ (a $Q \times M$ sparse matrix) by the $M$-dimensional vector $\pi_M^{(t)}$, the computational cost of this product is proportional to the number of non-zero elements in $P_{M \to Q}$ plus the dimension of the output vector, i.e., $O(\text{nnz}(P_{M \to Q}) + Q)$. Similarly, the cost of the transposed product is $O(\text{nnz}(P_{Q \to M}) + M)$.

The sum of non-zero elements across both original transition matrices, $\text{nnz}(P_{Q \to M}) + \text{nnz}(P_{M \to Q})$, represents the total number of correct and incorrect answers, which sum to $QM$. Specifically, $\text{nnz}(P_{Q \to M})$ is the total number of entries where $A_{ij} = 1$, and $\text{nnz}(P_{M \to Q})$ is the total number of entries where $\hat{A}_{ji} = 1$ (i.e., $A_{ij} = 0$).

Additional per-iteration operations, such as vector additions and scalar multiplications for incorporating the damping factor $\alpha$ and the uniform teleportation term, contribute $O(Q + M)$.

Combining these, the total complexity for one iteration is: $O(QM + Q + M) = O(QM)$

In typical scenarios the $QM$ term dominates $Q + M$. Therefore, the per-iteration complexity is effectively $O(QM)$. Thus, each iteration runs in $O(QM)$ time. Combining the number of iterations $T$ and the per-iteration complexity, the overall time complexity of EIP is:

$$O(T \cdot (QM + Q + M)) = O(\log(1/\epsilon) \cdot (QM + Q + M)) = O(tQM) \qquad (8)$$

Where $\epsilon$ is the predefined tolerance value, and $t$ is the convergence iteration count.

# E  ROBUSTNESS ANALYSIS

## E.1  ROBUSTNESS TO SINGLE/GROUP MODEL ADDITION

To assess EIP's ranking stability as the model pool composition evolves, simulating the integration of new models, we analyze its performance under perturbations. We consider scenarios analogous to incorporating a single new model or a small group of 5 new models into an existing evaluation set. This is achieved by examining the Spearman correlation ($\rho$) of question difficulty and model competency scores derived from reduced model pools (N-1 models via Leave-One-Out, and N-5 models via random subset removal) against the scores from the original full pool of N models. High correlations indicate that EIP's assessments remain consistent. The statistics for the N-5 scenario reflect averages over 10 trials. Results are summarized in Table 9.

Table 9: Comparison of EIP robustness: Leave-One-Out (LOO) vs. Random Removal of 5 Models. LOO was conducted for each model in the pool (30 trials). Random removal was conducted for 10 trials, each removing 5 models. Values are averaged Spearman correlation ($\rho$) over trials.

|  | LOO (Remove 1) | Random Removal (Remove 5) |
| --- | --- | --- |
| **Question Difficulty Correlation** |  |  |
| Mean Spearman $\rho$ | $0.9978 \pm 0.0024$ | $0.9863 \pm 0.0076$ |
| Max Observed $\rho$ Drop | 0.0100 | 0.0271 |
| **Model Competency Correlation** |  |  |
| Mean Spearman $\rho$ | 0.9998 | 0.9987 |
| **Question Set Reduction** |  |  |
| Mean Questions Removed | 32.9 | 217.9 |
| Max Questions Removed | 103 | 307 |
| Mean % Reduction | 0.09% | 0.62% |

The analysis presented in Table 9 underscores EIP's robust stability when its model pool composition evolves, akin to integrating new models.

**Model Competency rank exhibited greater stability:** The mean Spearman correlation was exceptionally high, starting at 0.9997 (SD=0.0002) with one model removed and remaining at 0.9934 (SD=0.0069) even with 15 models removed. This highlights EIP's capability to maintain consistent relative model rankings despite considerable variations in the composition and size of the model evaluation pool. A more intuitive visualization is shown in Figure 10.

**EIP maintains high consistency in question difficulty rankings despite significant model pool variations.** When simulating the addition of a single model (N-1 pool correlated with the N-model pool), the Spearman correlation ($\rho$) for question difficulty remains exceptionally high at $0.9978 \pm 0.0024$. Even with a more substantial change, such as integrating a group of 5 models (N-5 pool correlated with N-model pool, a 17% pool size change), the correlation for question difficulty remains very strong at $0.9863 \pm 0.0076$. This indicates that the relative difficulty assessment of questions is largely preserved.

**Model pool perturbations cause minimal changes to the effective question set.** EIP filters out questions universally solved or failed by the active model pool, and Table 9 shows that adding new models (simulated via LOO or random removal) has limited effect on this filtering. Even when integrating five models, the average change was only 217.9 questions (about 0.62% of the total), with a maximum of 307. This negligible impact ensures that nearly all questions continue contributing to the evaluation, and the exclusion of this small fraction does not disrupt the overall difficulty rankings, which remain highly stable.

When viewed from the perspective of an evolving model pool, the system demonstrates strong resilience, ensuring its evaluations are dependable as new models are incorporated into the assessment framework, with model competency rankings being particularly robust.

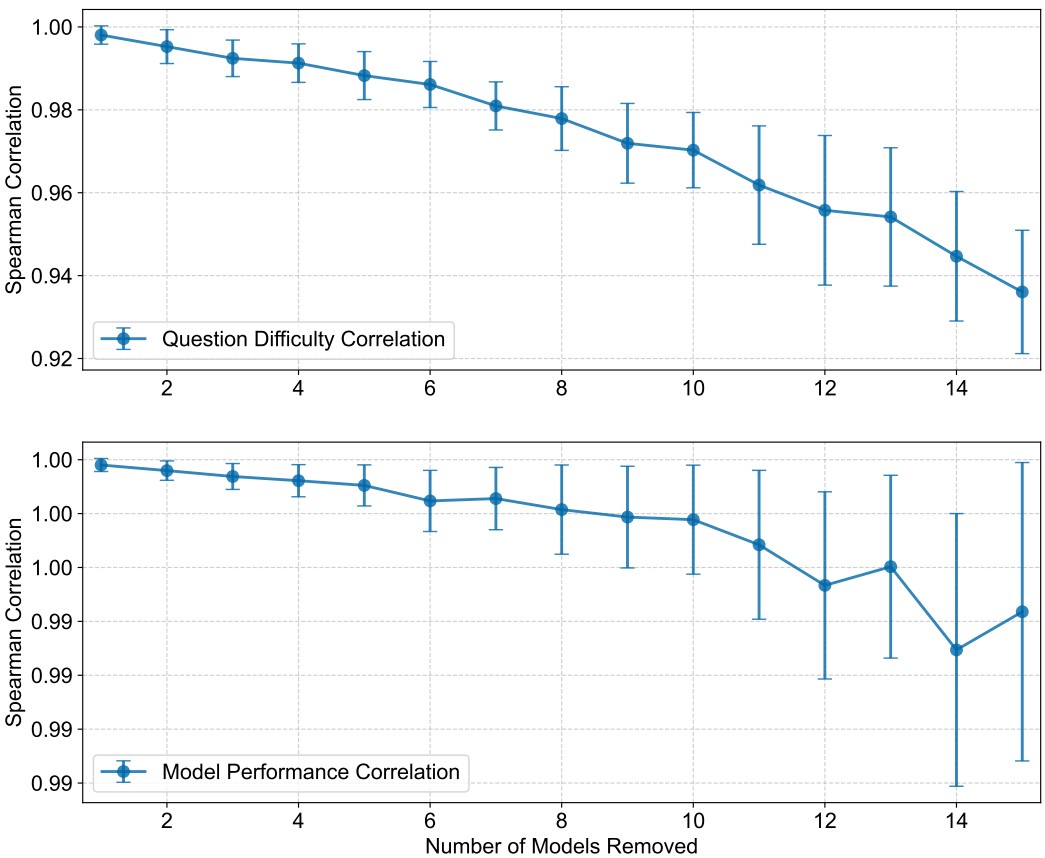

Figure 10: Impact of Random Model Subset Removal on EIP Stability. Top panel shows $\rho$ for question difficulty, and the bottom panel shows $\rho$ for model competency rank, both comparing results from the reduced model pool to the full model pool as $k$ models are removed. Error bars represent the standard deviation over multiple random removal trials for each $k$.

### E.2 Robustness to Dataset-Level Perturbation

To assess the influence of adding individual benchmark datasets on the final model rankings, we performed a leave-one-dataset-out analysis. The results are shown in Table 10.

Table 10: Robustness of EIP model competency rankings to dataset removal. Spearman $\rho$ correlation is calculated between rankings obtained after removing a dataset and the original rankings based on all datasets.

| Dataset Removed | # Questions | Model Rank Correlation ($\rho$) |
|---|---|---|
| HellaSwag | 10 042 | 0.9604 |
| MMLU-Pro | 12 032 | 0.9626 |
| MATH | 5000 | 0.9884 |
| GPQA | 646 | 0.9973 |
| GSM8k | 1319 | 0.9973 |
| BBH | 6511 | 0.9973 |
| **Average Correlation** | | 0.9839 |
| **Standard Deviation** | | 0.0162 |

**EIP model competency rankings exhibit high overall stability across all different dataset combinations.** The high average Spearman correlation between model rankings derived from N-1 datasets and the full N-dataset ranking($0.9839 \pm 0.0162$) indicates that the relative model hierarchy established by EIP is largely consistent, even when any single dataset is newly introduced to or excluded from the evaluation pool. When simulating the addition of substantial datasets like MMLU-Pro ($12,032$ questions) or HellaSwag ($10,042$ questions) to the remaining pool, the model rank correlations were observed to be more perturbed compared to adding smaller datasets. However, the correlations in these scenarios ($0.9604/0.9626$) still indicate a strong preservation of the relative model ranking.

These findings suggest that EIP provides model evaluations that are robust to variations in the specific suite of benchmark datasets used, irrespective of whether a large, broad dataset or a smaller, more specialized one is being integrated. This consistency points to its utility in generating more generalized assessments of model capabilities, less dependent on the idiosyncrasies of individual datasets.

## F   Answer Distribution Across Difficulty

As shown in Figure 11 (Highlighted in square box), a similar pattern emerged aligned with the case study in section 4: although GPT-4o answered slightly more questions correctly overall, Gemini-1.5-Pro outperformed it on difficult questions, achieving a 0.6% higher accuracy rate in this category. Consequently, despite answering 2.9% fewer medium-difficulty questions and achieving a lower overall accuracy rate, Gemini-1.5-Pro scored higher than GPT-4o under EIP, further emphasizing the impact of performance on high-difficulty questions. This real-world example corroborates the findings of our simulation, demonstrating EIP's ability to differentiate models based on their performance on challenging tasks.

## G   Analysis on Correlation and Difference between EIP and Accuracy-Based Model Rankings

While EIP's competency scores exhibit a strong overall correlation with traditional accuracy-based rankings (Kendall's Tau $\tau = 0.876$, $p < 0.001$, as detailed in Table 11), a closer examination reveals significant and insightful differences in how individual models and groups of models are ordered. This section delves into these inter-model ranking variations, underscoring EIP's ability to provide a more nuanced perspective on model capabilities than accuracy alone. As summarized in Table 11, the mean absolute rank change between the two methods is 1.60, with a median change of 1.0 rank position. Crucially, the maximum observed rank change for a model is 4 positions, indicating that for

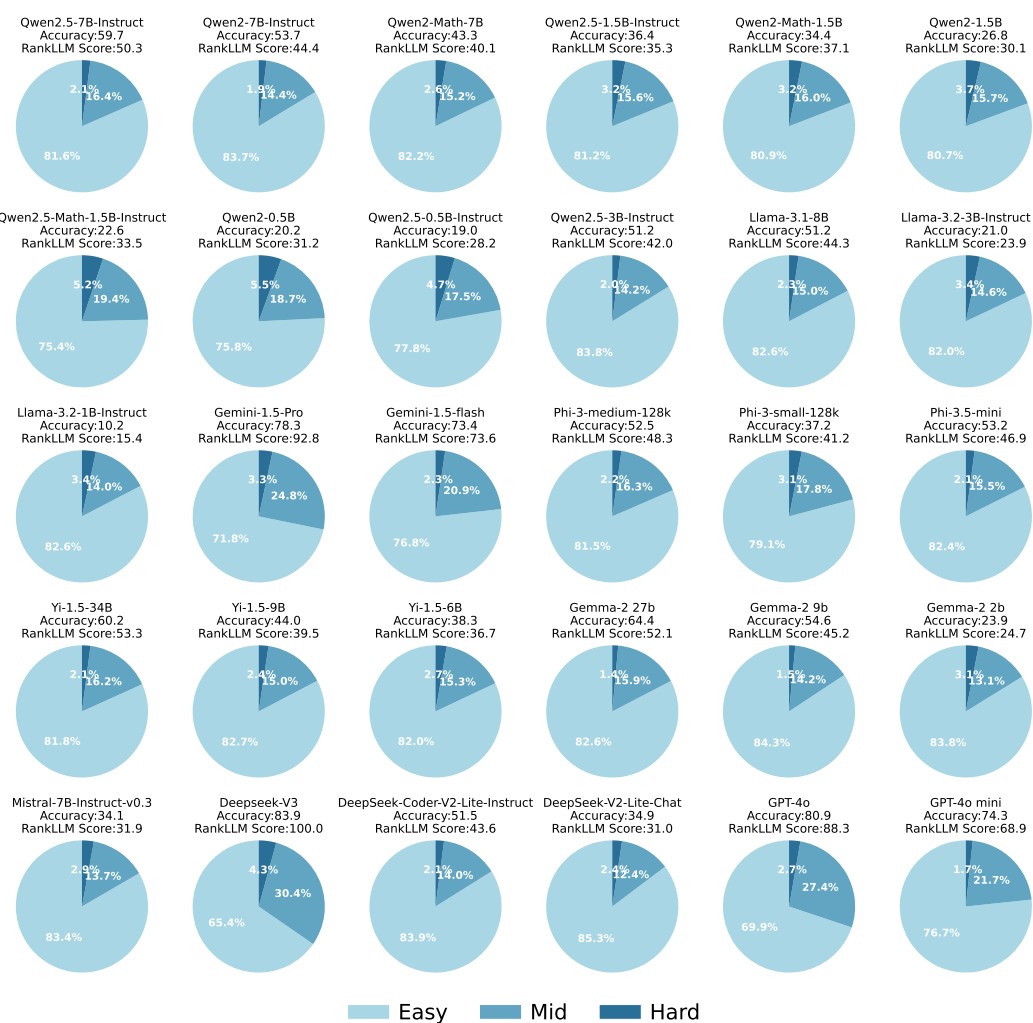

Figure 11: Distribution of correctly answered questions by models across the whole dataset (leave-one-out).

Table 11: Summary of model ranking comparison metrics between EIP and accuracy.

| Metric | Value |
| --- | --- |
| Mean Absolute Rank Change | 1.60 |
| Median Rank Change | 1.00 |
| Max Rank Change | 4.00 |
| Kendall's Tau Correlation | 0.876(p<0.001) |
| Rank-Biased Overlap (RBO) | 0.896 |
| ICC1 (Absolute Agreement) | 0.974 |

some models, the evaluation outcome under EIP can be substantially different from an accuracy-only assessment. The high Intraclass Correlation Coefficient (ICC1 = 0.974) suggests strong overall agreement in the rankings, yet the rank changes highlight the specific instances where EIP provides a distinct evaluation.

## G.1 DISTRIBUTION OF MODEL RANK CHANGES

The distribution of absolute rank changes (see Figure 12) reveals that while a majority of models experience small shifts (e.g., 6 models, or 20%, have no rank change), a notable portion shift by 1 or more positions. These larger shifts are particularly interesting as they point to models whose performance on questions of varying difficulty disproportionately affects their EIP score compared to their simple accuracy.

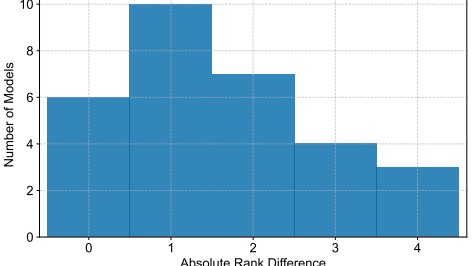

Figure 12: Absolute model rank difference distribution.

## G.2 LOCAL RANK DISPLACEMENT WITHIN ACCURACY-DEFINED TIERS

To further investigate where these ranking differences are most pronounced, we analyzed the local rank displacement of models within windows defined by their accuracy ranks. Figure 13a, Figure 13b, Figure 13c, and Figure 13d illustrate the mean and maximum rank displacement for models when grouped into tiers by their accuracy ranking, using window sizes of 1, 3, 5, and 10 respectively.

The trends observed in Figure 13 indicate varying degrees of rank displacement across accuracy-defined tiers:

**Relatively High Stability in Top-Tier Model Rankings.** For the top 10 models as ranked by accuracy (window "1-10" in Figure 13d), the mean displacement in EIP rank is comparatively low at 1.0, with a maximum displacement of 2.0. This suggests that among the highest-performing models by accuracy, EIP's rankings largely concur, although some re-ordering still occurs. The Kendall's Tau correlation within this specific window is very high ($0.867, p < 0.001$), indicating strong local rank agreement.

**Increased Rank Volatility in Mid-Tier Models.** The subsequent group of models, those ranked 11-20 by accuracy (window "11-20" in Figure 13d), shows increased rank displacement when evaluated by EIP. The mean displacement rises to 1.4, and the maximum displacement observed within this tier reaches 3.0. The local Kendall's Tau correlation ($0.733, p \approx 0.002$) remains statistically significant but is lower than that of the top tier, reflecting more substantial re-shuffling by EIP.

**Pronounced Rank Displacement in Lower-Tier Models.** Models in the lower third of the accuracy ranking (window "21-30" in Figure 13d) exhibit the highest mean displacement (2.4) and the overall maximum observed displacement of 4.0 rank positions. The local Kendall's Tau correlation drops further to $0.467$ ($p \approx 0.043$), indicating considerably weaker local rank agreement between accuracy and EIP in this tier. This suggests that for models with lower overall accuracy, EIP's difficulty-weighting mechanism has a more pronounced effect, leading to more substantial re-rankings based on performance on hard questions versus reliance on easier ones.

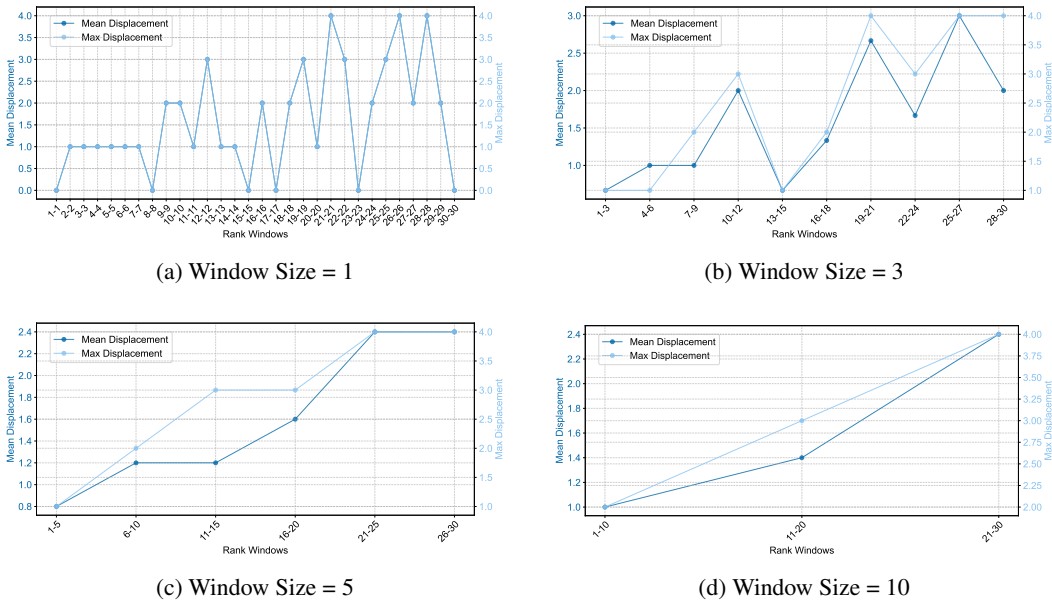

(a) Window Size = 1

(b) Window Size = 3

(c) Window Size = 5

(d) Window Size = 10

Figure 13: Mean and Maximum Rank Displacement of EIP scores compared to Accuracy scores within different Accuracy-Rank based windows. Windows are defined by accuracy rank (e.g., "1-10" refers to models ranked 1st to 10th by accuracy).

**Individual Model Rank Fluctuations Illustrate Local Reordering.** When examining displacements at the individual model level (window size 1, Figure 13a), fluctuations are evident across the spectrum. For instance, the model ranked 2nd by accuracy is displaced by 1 position in EIP, the model ranked 9th by accuracy is displaced by 2 positions, and models ranked 21st, 26th, and 28th by accuracy are all displaced by 4 positions in their respective EIP rankings. This highlights that even when broader trends might align, EIP provides distinct local orderings based on its difficulty-aware assessment.

These windowed analyses demonstrate that while EIP broadly agrees with accuracy at the very top of the performance spectrum, its differentiation based on question difficulty becomes more apparent and impactful in the middle and lower tiers of accuracy rankings. This is where the ability to correctly answer challenging questions, or the failure to do so, most significantly revises a model's perceived competency compared to a simple count of correct answers. The maximum displacement of 4 rank positions underscores that EIP can lead to meaningfully different conclusions about relative model strengths, particularly for models with nuanced performance profiles across varying question difficulties.

## H   EXPERIMENT DETAILS

To ensure optimal performance and compatibility across our experiments, we employed a combination of advanced software libraries and frameworks, including Together AI 1.2.1, OpenAI 1.30.3, vLLM 0.5.5, FlashInfer 0.1.5+cu124, Flash-Attn 2.6.3, Torch 2.4.0, Google Generative AI 0.7.2, torch 2.5.1 and CUDA 12.4.

### H.1   DATASET & PROMPT.

We adhered to the standard configurations outlined in the respective original benchmarks used in our experiments. For the benchmarks BBH, MMLU-Pro, GSM8k, GPQA, MATH, and HellaSwag, we utilized 5-shot prompting combined with Chain of Thought (CoT) reasoning. These strategies were chosen to align with the original settings for each respective benchmark. Temperature was set to 0 to ensure reproducibility.

## H.2 Model Abbreviation.

For simplicity, we use some model abbreviations in figures and tables. Specifically, DeepSeek-Coder-Lite is DeepSeek-Coder-V2-Lite-Instruct, Qwen2.5-3B is Qwen2.5-3B-Instruct, Qwen2.5-1.5B is Qwen2.5-1.5B-Instruct, Qwen2.5-Math-1.5B is Qwen2.5-Math-1.5B-Instruct, Mistral-7B-v0.3 is Mistral-7B-Instruct-v0.3, DeepSeek-Chat-Lite is DeepSeek-V2-Lite-Chat, Qwen2.5-0.5B is Qwen2.5-0.5B-Instruct, Llama-3.2-3B is Llama-3.2-3B-Instruct, Llama-3.2-1B is Llama-3.2-1B-Instruct.

Table 12: Information on the selected models used in EIP.

| Model Name | Size | Developer | Version | Context Length | Open Weight |
|---|---|---|---|---|---|
| ChatGPT-4o (OpenAI et al., 2024a) | N/A | OpenAI | Aug 2024 | 128K | ✗ |
| ChatGPT-4o-mini (OpenAI et al., 2024c) | N/A | OpenAI | July 2024 | 128K | ✗ |
| DeepSeek-V3(DeepSeek-AI et al., 2024a) | 671B | DeepSeek | Dec 2024 | 64K | ✓ |
| DeepSeek-Coder-V2-Lite-Instruct(DeepSeek-AI et al., 2024b) | 15.7B | DeepSeek | June 2024 | 32K | ✓ |
| DeepSeek-V2-Lite-Chat(DeepSeek-AI et al., 2024b) | 15.7B | DeepSeek | May 2024 | 32K | ✓ |
| Gemini-1.5-Pro(Team et al., 2024a) | N/A | Google | May 2024 | 2M | ✗ |
| Gemini-1.5-Flash(Team et al., 2024a) | N/A | Google | May 2024 | 1M | ✗ |
| Gemma-2-27b-it(Team et al., 2024b) | 27.2B | Google | June 2024 | 8K | ✓ |
| Gemma-2-9b-it(Team et al., 2024b) | 9.B | Google | June 2024 | 8K | ✓ |
| Gemma-2-2b-it(Team et al., 2024b) | 2.2B | Google | June 2024 | 8K | ✓ |
| Llama-3.1-8B-Instruct(Grattafiori et al., 2024) | 8.03B | Meta | July 2024 | 128K | ✓ |
| Llama-3.2-3B-Instruct(Grattafiori et al., 2024) | 3.21B | Meta | Sep. 2024 | 128K | ✓ |
| Llama-3.2-1B-Instruct(Grattafiori et al., 2024) | 1.24B | Meta | Sep. 2024 | 128K | ✓ |
| Mistral-7B-Instruct-v0.3(Jiang et al., 2023) | 7.25B | Mistral AI | May 2024 | 32K | ✓ |
| Phi-3-medium-128k-instruct(Abdin et al., 2024) | 14B | Microsoft | May 2024 | 128K | ✓ |
| Phi-3-small-128k-instruct(Abdin et al., 2024) | 7.39B | Microsoft | May 2024 | 128K | ✓ |
| Phi-3.5-mini-128k-instruct(Abdin et al., 2024) | 3.82B | Microsoft | Aug 2024 | 128K | ✓ |
| Qwen2-7B-Instruct (Yang et al., 2024) | 7.62B | Qwen | June 2024 | 131K | ✓ |
| Qwen2-Math-7B-Instruct (Yang et al., 2024) | 7.62B | Qwen | June 2024 | 131K | ✓ |
| Qwen2-1.5B-Instruct (Yang et al., 2024) | 1.54B | Qwen | June 2024 | 131K | ✓ |
| Qwen2-Math-1.5B-Instruct (Yang et al., 2024) | 1.54B | Qwen | June 2024 | 131K | ✓ |
| Qwen2-0.5B-Instruct (Yang et al., 2024) | 0.49B | Qwen | June 2024 | 32K | ✓ |
| Qwen2.5-7B-Instruct (Yang et al., 2024) | 7.62B | Qwen | Sep. 2024 | 131K | ✓ |
| Qwen2.5-Math-7B-Instruct (Yang et al., 2024) | 7.62B | Qwen | Sep. 2024 | 131K | ✓ |
| Qwen2.5-1.5B-Instruct (Yang et al., 2024) | 1.54B | Qwen | Sep. 2024 | 131K | ✓ |
| Qwen2.5-Math-1.5B-Instruct (Yang et al., 2024) | 1.54B | Qwen | Sep. 2024 | 131K | ✓ |
| Qwen2.5-3B-Instruct (Yang et al., 2024) | 3.09B | Qwen | Sep. 2024 | 131K | ✓ |
| Yi-1.5-34B-Chat (AI et al., 2025) | 34.4B | 01-AI | May 2024 | 16K | ✓ |
| Yi-1.5-9B-Chat (AI et al., 2025) | 8.83B | 01-AI | May 2024 | 16K | ✓ |
| Yi-1.5-6B-Chat (AI et al., 2025) | 6.06B | 01-AI | May 2024 | 16K | ✓ |

## H.3 Evaluation Protocol

To ensure robust human evaluations, we implemented two key design principles: 1) Discipline selection: Varied domains (e.g., mathematics, computer science, commonsense reasoning) with matched pair disciplines. 2) Blind judgments: Evaluators were unaware of model or peer judgments.

Each of the 20 evaluators assessed 70 randomly ordered question pairs via a verification interface (Appendix P) to mitigate order bias. We computed two primary alignment metrics: 1) Individual Alignment: For each evaluator, the percentage of their non-skipped judgments ($V_1$–$V_{20}$) aligning with method predictions. 2) Consensus Alignment: The proportion of questions where a method's prediction matches the majority of human judgment. For the set of non-skipped questions $Q$, this is defined as:

$$\text{Consensus} = \frac{1}{|Q|} \sum_{q \in Q} \mathbb{I}\left(\text{pred}_q = \arg\max_{j \in \mathcal{V}_q} \text{count}_q(j)\right) \qquad (9)$$

**Evaluator Information.** Our human evaluation involved a total of 20 participants. This pool included the two authors and 18 current students not holding terminal degrees, ranging from associate-level to PhD candidates. To ensure a diverse range of perspectives, participants were recruited from various academic backgrounds, and gender balance was also considered. All evaluators possessed strong English proficiency, enabling effective task completion. Table 13 summarizes the academic fields and educational levels of the 18 student evaluators.

Evaluators with backgrounds in computer science, mathematics, and engineering had all completed formal coursework in calculus, probability, and statistics, providing them with solid quantitative foundations. In contrast, participants from the arts, social sciences, and humanities may not have

Table 13: Diversity of Human Evaluator Pool (N=18, excluding authors).

| Academic Field | Undergraduate | Graduate (Master's) | PhD Candidate | Associate Degree | Total by Field |
|---|---|---|---|---|---|
| Computer Science | 4 | 1 | 1 | 0 | 6 |
| Other Engineering | 2 | 1 | 0 | 0 | 3 |
| Arts | 1 | 1 | 0 | 0 | 2 |
| Business | 1 | 1 | 0 | 0 | 2 |
| Humanities | 3 | 0 | 0 | 1 | 4 |
| Social Sciences | 2 | 0 | 0 | 1 | 3 |
| **Total by Level** | **13** | **4** | **1** | **2** | **20 (incl. 2 authors)** |

received such training. To accommodate differences in subject knowledge, participants were allowed to voluntarily skip domain-specific questions if they felt unqualified to answer them. This mechanism helped minimize the influence of unfamiliar content on the evaluation process.

Domain-specific questions requiring technical expertise were handled by participants with the appropriate background, while common sense reasoning tasks were completed by all evaluators. This design ensured that our human evaluation captured both specialized capabilities and general reasoning performance across a diverse participant pool.

### H.4 IRT BASELINE CONFIGURATIONS

To provide a robust comparison, we implemented and configured three Item Response Theory (IRT) baseline models. These configurations, detailed below, were chosen to give the IRT models ample opportunity to converge and perform optimally, even on the relatively small 100-question dataset used in our controlled simulation (section 4).

**1PL IRT (Rasch Model) and 2PL IRT**: Both the 1-Parameter Logistic (1PL) IRT model, also known as the Rasch model (where item discrimination is fixed at 1), and the 2-Parameter Logistic (2PL) IRT model were fitted by maximizing the log-likelihood of the observed response matrix. Optimization was performed using the L-BFGS-B algorithm, a quasi-Newton method suitable for bounded parameter estimation. A maximum of 1000 iterations was permitted for the optimization process. To enhance parameter stability and prevent extreme values, L2 regularization with a coefficient of $0.01$ was applied to both model abilities ($\theta$) and item difficulties ($\beta$). For the 2PL model, L2 regularization was additionally applied to the item discrimination parameters ($\alpha$) centered around 1 (i.e., penalizing $(\alpha - 1)^2$), encouraging discriminations to be in a standard range. Parameter bounds were enforced during optimization: abilities and difficulties were constrained to the interval $[-5, 5]$, and discrimination parameters for the 2PL IRT were constrained to $[0.1, 5]$ to ensure positivity and interpretability.

**Multidimensional IRT (Multi-IRT)**: We implemented a Multidimensional IRT (MIRT) model configured with $D = 3$ latent dimensions. This choice was made to explore a more complex latent trait structure, though we acknowledge that estimating parameters for three dimensions from a 100-question dataset can be challenging. The MIRT model was trained for 1000 epochs using the Adam optimizer with a learning rate of $0.01$. Parameter estimation employed variational inference (VI), where the true posterior distributions of abilities, difficulties, and discriminations were approximated. Standard Normal distributions, $\mathcal{N}(0, 1)$, served as priors for all model parameters (abilities, difficulties, and per-dimension discrimination values). For the variational posterior approximation, we utilized a mean-field approach where the posterior for each parameter was modeled as a Normal distribution with a learnable mean and a fixed standard deviation of 1. The model was trained by maximizing the Evidence Lower Bound (ELBO). The reported ability scores for Multi-IRT are the L2 norm of the per-model multidimensional ability vectors, subsequently min-max scaled to a 0-100 range, similar to other IRT models and EIP scores for comparability.

## I SENSITIVITY ANALYSIS OF THE DAMPING FACTOR

The damping factor $\alpha$ in EIP (cf. Equation 2–Equation 3) controls the strength of teleportation regularization and influences the smoothness of the stationary distributions over questions and models. To verify that our conclusions are not sensitive to this choice, we recomputed EIP on the full MetaBench response matrix (30 models $\times$ 35,550 questions) across a broad grid of $\alpha$ values and compared the induced model- and question-rankings against a reference run with $\alpha = 1.00$ using Spearman correlation. Table 14 shows that both model and question rankings remain highly stable across this range.

Table 14: Sensitivity of EIP rankings to the damping factor $\alpha$. We report Spearman correlations of model and question rankings relative to $\alpha = 1.00$ on the full MetaBench matrix.

| $\alpha$ | $\rho_{\text{model}}$ | $\rho_{\text{question}}$ |
|---|---|---|
| 0.50 | 0.9911 | 0.9976 |
| 0.60 | 0.9942 | 0.9985 |
| 0.70 | 0.9978 | 0.9992 |
| 0.85 | 0.9987 | 0.9998 |
| 0.95 | 1.0000 | 1.0000 |
| 0.99 | 1.0000 | 1.0000 |

Even under a conservative setting (e.g., $\alpha = 0.50$), model rankings remain highly stable ($\rho_{\text{model}} = 0.9911$) and question rankings are even more stable ($\rho_{\text{question}} = 0.9976$). For $\alpha \geq 0.95$, the induced rankings are effectively identical ($\rho = 1.0000$ for both models and questions). In all experiments in the main paper, we fix $\alpha = 0.85$, which balances fast geometric convergence with a sufficiently expressive stationary distribution (see Table 8).

## J ROBUSTNESS TO WEAK MODEL INFLATION

A natural concern is that expanding the evaluation pool with many weak models could inflate question difficulty scores and perturb rankings. EIP mitigates this effect by attributing a failing model's marginal contribution to question difficulty proportionally to $\pi_m / F(m)$, where $\pi_m$ is the model competency score and $F(m)$ is its total number of failures; weak models tend to have both small $\pi_m$ and large $F(m)$, limiting their per-question influence. We stress-test this scenario by duplicating the 10 weakest models (two identical copies each), expanding the pool from 30 to 50 models, and recomputing EIP on the full response matrix.

Table 15: Weak-model inflation stress test (30 $\rightarrow$ 50 models by adding 20 weak copies). We compare question difficulties (Pearson) and the ranking of the original 30 models (copies excluded).

| Statistic | Value |
|---|---|
| Pearson $r$ (question difficulties; 30 vs. 50 models) | 0.9972 |
| Spearman $\rho$ (model ranks; original 30 models) | 0.9969 |
| Unchanged in Top-20 models (by rank) | 20/20 |
| Max absolute rank change among Bottom-10 models | 2 |

Table 15 confirms that both the question-difficulty landscape and the induced model ranking are highly stable under this extreme weak-model inflation: the Top-20 models remain unchanged, and the Bottom-10 models shift by at most two positions. To further illustrate why weak models do not dominate difficulty estimates, Table 16 provides a concrete per-question contribution breakdown. For question 14703, 15 models answer incorrectly; GPT-4o alone accounts for 32.69% of the wrong-side contribution mass, whereas the 7 weakest failing models combined contribute only 18.60%.

## K FAMILY-LEVEL BLIND SPOT ANALYSIS

Another concern is whether "family-level blind spots" could systematically bias difficulty estimates, e.g., if a single architecture family disproportionately fails a set of questions and thereby dominates

Table 16: Wrong-side contribution breakdown for question 14703. For each failing model $m$, we report its EIP competency score $\pi_m$ (max-normalized to 100), its total failure count $F(m)$ over the 35,550-question matrix, and its percentage share of the question difficulty, proportional to $\pi_m/F(m)$ among failing models.

| Model | Model score $\pi_m$ | # wrong $F(m)$ | % of difficulty |
|---|---|---|---|
| GPT-4o | 88.3262 | 7308 | 32.69% |
| Yi-1.5-34B | 53.2638 | 14518 | 9.92% |
| Qwen2.5-7B-Instruct | 50.2553 | 14699 | 9.25% |
| Phi-3-medium-128k | 48.3485 | 17207 | 7.60% |
| Qwen2-7B | 44.4336 | 16800 | 7.15% |
| Qwen2-Math-7B | 40.1448 | 20416 | 5.32% |
| Yi-1.5-9B | 39.4718 | 20176 | 5.29% |
| Qwen2.5-1.5B-Instruct | 35.2976 | 22845 | 4.18% |
| Mistral-7B-Instruct-v0.3 | 31.8986 | 23617 | 3.65% |
| Qwen2.5-Math-1.5B | 33.5447 | 27613 | 3.29% |
| Qwen2-0.5B | 31.1981 | 28460 | 2.96% |
| Qwen2.5-0.5B-Instruct | 28.2489 | 28859 | 2.65% |
| gemma-2-2b | 24.7172 | 27192 | 2.46% |
| Llama-3.2-3B-Instruct | 23.8847 | 28205 | 2.29% |
| Llama-3.2-1B-Instruct | 15.3590 | 31967 | 1.30% |

their difficulty mass. We quantify this effect by aggregating each question's wrong-side contributions by model family (Qwen, Llama, Yi, etc.) and marking a question as *family-dominated* when a single family accounts for more than 90% of the wrong-side contribution mass and has at least two failing members on that question. Table 17 shows that such extreme cases are rare in our diverse 30-model pool: only 246 out of 35,550 questions (0.69%) are family-dominated. Moreover, none of these dominated questions appear among the globally hardest items (none are in the Top-1,000 hardest questions), and their difficulty scores fall in the moderate-to-easy range (median 11.73; maximum 23.07 on the EIP question scale).

Table 17: Family-dominated questions under a 90% dominance threshold (and requiring at least two failing models within the dominating family).

| Family | # dominated questions | Share of all questions |
|---|---|---|
| Qwen | 232 | 0.65% |
| Llama | 13 | 0.04% |
| Phi | 1 | 0.00% |
| **Total** | **246** | **0.69%** |

## L    ROBUSTNESS TO SUPER MODEL ADDITION

To further probe robustness to pool perturbations, we simulate adding a hypothetical "super model" that answers a large fraction of questions correctly (99%, 95%, 90%, or 80% accuracy) and recompute EIP on the expanded pool. Table 18 reports that, after excluding the added synthetic model, the Spearman correlation between the original 30-model ranking and the post-addition ranking remains extremely high ($\rho_{\text{model}} \in [0.9956, 0.9973]$), with a maximum absolute rank change of 2 across all settings.

## M    EXTENDED COMPUTATIONAL EFFICIENCY ANALYSIS

We provide an extended computational-efficiency comparison for EIP and IRT baselines on the full MetaBench scale (30 models $\times$ 35,550 questions). Table 19 reports wall-clock runtimes on consumer CPUs and high-performance GPUs. EIP's update is dominated by sparse matrix–vector multiplications and converges in 9 iterations (see Appendix C), resulting in millisecond-level total

Table 18: Robustness to adding a hypothetical super model. We append a synthetic model and recompute EIP; correlations and rank changes are computed on the original 30 models (excluding the added model).

| Super model accuracy | $\rho_{\mathrm{model}}$ | Mean $|\Delta\mathrm{rank}|$ | Max $|\Delta\mathrm{rank}|$ |
|---|---|---|---|
| 99% | 0.9964 | 0.40 | 2 |
| 95% | 0.9956 | 0.47 | 2 |
| 90% | 0.9973 | 0.33 | 2 |
| 80% | 0.9973 | 0.33 | 2 |

Table 19: Computational efficiency comparison on the full response matrix (30 models $\times$ 35,550 questions). EIP totals correspond to 9 iterations until convergence; IRT runtimes report end-to-end fitting times for 1PL and 2PL baselines.

| Hardware | EIP (1 iter.) | EIP (total) | 1PL-IRT | 2PL-IRT |
|---|---|---|---|---|
| Intel i7 CPU | 0.00597s | 0.05373s | 1782.75s | 3787.03s |
| MPS (Mac M1) | 0.012296s | 0.110804s | – | – |
| A40 GPU (CUDA) | 0.000581s | 0.005263s | 24.09s | 459s |
| B200 GPU (CUDA) | 0.000510s | 0.004634s | 5.33s | 0.59s |

runtime even on CPU; GPU acceleration further reduces the runtime. In contrast, IRT requires iterative likelihood maximization and is orders of magnitude slower at this scale.

# N    ANCHOR QUESTION SAMPLING FOR RAPID EVALUATION

Finally, we study whether EIP rankings can be recovered from a small subset of questions, which is relevant to adaptive or budgeted evaluation settings. We perform difficulty-stratified sampling (deciles by EIP difficulty) and select only 0.5%, 1%, or 5% of questions as anchors, recompute EIP on the anchor-only matrix, and compare the resulting model ranking against the full-question ranking. Table 20 shows that even with 0.5% anchors (180 questions), the induced model ranking remains strongly correlated with the full ranking ($\rho_{\mathrm{model}} = 0.8901$), improving further as the anchor budget increases.

Table 20: Difficulty-stratified anchor sampling for rapid evaluation. We recompute EIP using only the sampled anchors and compare the resulting model ranking against the full-question ranking.

| Anchor rate | # anchors | $\rho_{\mathrm{model}}$ | Avg $|\Delta\mathrm{rank}|$ |
|---|---|---|---|
| 0.5% | 180 | 0.8901 | 3.07 |
| 1% | 360 | 0.9199 | 2.47 |
| 5% | 1780 | 0.9729 | 1.40 |

# O    ASSET LICENSING AND TERMS OF USE

This appendix details the licensing information for all software libraries, datasets, benchmarks, and language models utilized in the experiments presented in this paper. Ensuring compliance with the respective terms of use is critical for reproducible and ethical research.

## O.0.1    SOFTWARE LIBRARY LICENSES

The software libraries employed in this research are governed by a variety of open-source licenses, predominantly permissive ones, which facilitate their use in academic and research settings. Table 21 provides a summary.

**Notes on Software Library Licenses:**    **Together AI Python SDK 1.2.1** is licensed under Apache 2.0, as confirmed by its official repository. Community SDKs may vary. **OpenAI Python Library**

Table 21: Software Library Licenses

| Library | Version | License |
|---|---|---|
| Together AI Python SDK | 1.2.1 | Apache License 2.0 |
| OpenAI Python Library | 1.30.3 | Apache License 2.0 |
| vLLM | 0.5.5 | Apache License 2.0 |
| FlashInfer | 0.1.5+cu124 | Apache License 2.0 |
| Flash-Attn | 2.6.3 | BSD 3-Clause |
| Torch (PyTorch) | 2.4.0 | BSD 3-Clause |
| Google Generative AI SDK | 0.7.2 | Apache License 2.0 |
| torch (PyTorch) | 2.5.1 | BSD 3-Clause |
| NVIDIA CUDA Toolkit | 12.4 | NVIDIA CUDA Toolkit EULA |

**1.30.3** is licensed under Apache 2.0 according to its official repository, though older PyPI versions might indicate MIT; the current repository is considered authoritative. **vLLM 0.5.5** is licensed under Apache 2.0, as per its PyPI page and official repository. **FlashInfer 0.1.5+cu124** is licensed under Apache 2.0, with the "+cu124" denoting CUDA 12.4 compatibility. **Flash-Attn 2.6.3** is licensed under BSD 3-Clause, according to its official repository. **PyTorch (Torch 2.4.0 & 2.5.1)** is licensed under BSD 3-Clause, as confirmed by its official repository and PyPI for version 2.5.1. **Google Generative AI SDK 0.7.2** is licensed under Apache 2.0, as per its official repository. Newer developments may be in 'google-genai' under the same license. **NVIDIA CUDA Toolkit 12.4** is governed by the NVIDIA CUDA Toolkit EULA, a proprietary license permitting development and specific redistribution.

### O.0.2 DATASET AND BENCHMARK LICENSES

The datasets and benchmarks employed are governed by various open-source licenses or public domain dedications. Table 22 summarizes this information.

Table 22: Dataset and Benchmark Licenses

| Dataset/Benchmark | Stated License(s) |
|---|---|
| BBH (Big-Bench Hard) (Suzgun et al., 2022) | Apache 2.0 (for BIG-Bench (Srivastava et al., 2023)) / MIT (for specific BBH repository by Suzgun et al.) |
| MMLU-Pro (Wang et al., 2024) | Apache License 2.0 |
| GSM8k (Cobbe et al., 2021) | MIT License (likely, from original OpenAI repository) |
| GPQA (Rein et al., 2023) | MIT License |
| MATH (Hendrycks et al., 2021b) | MIT License |
| HellaSwag (Zellers et al., 2019) | MIT License (original author's repository) |

**Notes on Dataset and Benchmark Licenses:** **BBH** is derived from BIG-Bench (Srivastava et al., 2023), which is under Apache 2.0. The specific repository for BIG-Bench Hard by Suzgun et al. (2022) uses an MIT license. The source utilized determines the applicable license. **MMLU-Pro** is clearly licensed under Apache 2.0 by TIGER-AI-Lab (Wang et al., 2024). **GSM8k**, as originally released by OpenAI (Cobbe et al., 2021), is likely under an MIT License. It's important to note that variants like GSM8k-Platinum may use CC-BY-4.0; the specific source license is key. **GPQA** by Rein et al. (2023) is under an MIT License. The SuperGPQA variant uses ODC-BY. **MATH** dataset by Hendrycks et al. (2021b) is clearly MIT licensed. **HellaSwag**, from the original authors' repository (Zellers et al., 2019), is under an MIT License. Other platforms hosting HellaSwag (e.g., Kaggle: CC0; Hugging Face/jon-tow: CC BY NC 4.0) may have different licenses; the original MIT license was considered authoritative for the version used.

### O.0.3 LANGUAGE MODEL LICENSING AND TERMS OF USE

Language models are subject to API service agreements or specific open-weight licenses. Table 23 provides an overview.

Table 23: Language Model Licensing and Terms Overview

| Model Family/Provider | Primary Governing Terms |
|---|---|
| OpenAI (ChatGPT series, GPT-4o (OpenAI et al., 2024a), GPT-4o-mini (OpenAI et al., 2024b)) | OpenAI Usage Policies & Services Agreement |
| Google Gemini (API: Gemini-1.5-Pro, Gemini-1.5-Flash (Team et al., 2024a)) | Google APIs ToS & Gemini API Additional ToS |
| Google Gemma (Open Weights: Gemma-2 series (Team et al., 2024b)) | Gemma Terms of Use |
| Meta Llama (Llama 3.1, Llama 3.2 series (Grattafiori et al., 2024)) | Llama Community License Agreement & Acceptable Use Policy |
| DeepSeek AI (DeepSeek-V3 (DeepSeek-AI et al., 2024a), Coder-V2-Lite, V2-Lite-Chat (DeepSeek-AI et al., 2024b)) | DeepSeek Model License / MIT License (varies by model) |
| Mistral AI (Mistral-7B-Instruct-v0.3 (Jiang et al., 2023)) | Apache License 2.0 |
| Alibaba Qwen (Qwen2, Qwen2.5 series (Yang et al., 2024)) | Apache License 2.0 (most) / Qwen RESEARCH LICENSE (Qwen2.5-3B-Instruct) |
| 01.AI Yi (Yi-1.5 series (AI et al., 2025)) | Apache License 2.0 |
| Microsoft Phi (Phi-3, Phi-3.5 series (Abdin et al., 2024)) | MIT License |

**Notes on Language Model Licenses:** **OpenAI Models'** API use is governed by OpenAI's Usage Policies and Services Agreement. Customer Content (input/output for paid API tiers) is not used for training OpenAI models, as per their Services Agreement. **Google Gemini API** is governed by Google APIs Terms of Service and the Gemini API Additional Terms of Service. Data usage for improvement depends on the service tier, according to the Gemini API Terms of Service. **Google Gemma Open Weights** are governed by the Gemma Terms of Use, allowing modification and distribution with attribution and adherence to a Prohibited Use Policy. **Meta Llama Models** are released under version-specific Llama Community License Agreements, requiring attribution and adherence to an Acceptable Use Policy (AUP). Commercial use by entities with over 700 million monthly active users typically requires a separate license, as detailed in the Llama 3 Community License, for example. **DeepSeek AI Models'** code is often MIT licensed. Some model weights are also MIT (e.g., V3-0324 release), while others are under a custom DeepSeek Model License with use restrictions. The specific license for each model must be checked. **Mistral AI Models**, such as Mistral-7B-Instruct-v0.3 (Jiang et al., 2023), are available under Apache 2.0. Premier models may have different licenses. **Alibaba Qwen Models** are mostly Apache 2.0. However, specific versions like Qwen2.5-3B-Instruct use a non-commercial Qwen RESEARCH LICENSE AGREEMENT. The license for each specific model should be verified. **01.AI Yi Models'** open-source releases (AI et al., 2025) are under Apache 2.0. **Microsoft Phi Models**, such as those described by Abdin et al. (2024), are generally released under the permissive MIT License.

**Compliance Statement** To the best of the authors' knowledge, and based on the diligent research of the licenses and terms detailed herein, the use of all software, datasets, benchmarks, and language models in this study complies with their respective governing terms and conditions. This proactive approach to documenting and adhering to licensing requirements is fundamental to responsible and ethical AI research.

---

**Prompt O.1: Example Prompt of BBH, few-shot CoT Prompts are formed by original research team**

You are a logic expert, you are given questions that involve enumerating objects and asking the model to count them.

Example 1:
<Example Question 1>
Answer:
<Example Answer 1>

Example 2:
<Example Question 2>
Answer:
<Example Answer 2>

Example 3:
<Example Question 3>
Answer:
<Example Answer 3>

Example 4:
<Example Question 4>
Answer:
<Example Answer 4>

Example 5:
<Example Question 5>
Answer:
<Example Answer 5>

1. **Answer Formatting**:
- **Multiple Choice Questions with Options**: Only select from the provided options (e.g., A, B, C, or D). If you calculate a numerical answer (e.g., "10") that matches an option, respond with the corresponding option letter (e.g., "$\boxed{A}$"), **not** the number itself. Failing to select the option will be marked as incorrect.
- **Short Answer Questions**: Provide the final answer in the format "$\boxed{X}$", where X is the correct answer. Do not include any additional formatting or explanations inside "$\boxed{}$". Do not answer only the serial number as well, Remember if answer is 1.Henry, respond with "$\boxed{Henry}$" but not "$\boxed{1}$".

2. **Answer Example**:
- For multiple choice: If the calculated answer is 10 and "10" corresponds to option C, respond with "$\boxed{C}$".
- For math questions: If the correct answer is "42," respond with "$\boxed{42}$".
- For short answer question other than math, Remember if answer is 1.Henry, respond with "$\boxed{Henry}$" but not "$\boxed{1}$".

3. **Additional Instructions**:
- Place any reasoning or calculations outside of the "$\boxed{}$" notation if necessary.
- Use "$\boxed{}$" only for the final answer.

4. **Think step by step and answer carefully**.
- You must think before answering the question.
- You must answer step by step.
Question: <question>

**Prompt O.2: Example Prompt of GSM8k, CoT Prompts are sampled from original test set**

You are a math expert, and you are tasked with answering questions in math with example to reference.

Example 1:
<Example Question 1>
Answer:
<Example Answer 1>

Example 2:
<Example Question 2>
Answer:
<Example Answer 2>

Example 3:
<Example Question 3>
Answer:
<Example Answer 3>

Example 4:
<Example Question 4>
Answer:
<Example Answer 4>

Example 5:
<Example Question 5>
Answer:
<Example Answer 5>

You've finished reading all the examples
Now read the question carefully and answer according to the following guidelines:

1. **Answer Formatting**:
- **Multiple Choice Questions with Options**: Only select from the provided options (e.g., A, B, C, or D). If you calculate a numerical answer (e.g., "10") that matches an option, respond with the corresponding option letter (e.g., "$\boxed{A}$"), **not** the number itself. Failing to select the option will be marked as incorrect.
- **Short Answer Questions**: Provide the final answer in the format "$\boxed{X}$", where X is the correct answer. Do not include any additional formatting or explanations inside "$\boxed{}$". Do not answer only the serial number as well, Remember if answer is 1.Henry, respond with "$\boxed{Henry}$" but not "$\boxed{1}$".

2. **Answer Example**:
- For multiple choice: If the calculated answer is 10 and "10" corresponds to option C, respond with "$\boxed{C}$".
- For math questions: If the correct answer is "42," respond with "$\boxed{42}$".
- For short answer question other than math, Remember if answer is 1.Henry, respond with "$\boxed{Henry}$" but not "$\boxed{1}$".

3. **Additional Instructions**:
- Place any reasoning or calculations outside of the "$\boxed{}$" notation if necessary.
- Use "$\boxed{}$" only for the final answer.

4. **Think step by step and answer carefully**.
- You must think before answering the question.
- You must answer step by step.
Question: <question>

## P  VERIFICATION TEST EXAMPLE

---

**Prompt P.1: Verification Test Example**

Welcome to EIP Test Program!

Please enter your First Name for identification:

===== Now at Group 1 / 100 (Group ID: 0) =====

Progress: [————————————————] 0.0%

1. Question:
Evaluate the expression
$(751 - 745) + (748 - 742) + (745 - 739) + (742 - 736) + \cdots + (499 - 493) + (496 - 490).$

2. Question:
What is the value of the inflection point of $f(x) = \frac{10 \ln x}{x^2}$?

A.  2.000   B.  1.587   C.  0.693   D.  1.203   E.  3.014   F.  2.718   G.  4.000
H.  3.142   I.  1.000   J.  2.301

Which question is more difficult? Enter 0 if unable to judge, otherwise enter 1 or 2:

---

## Q    DISCLOSURE OF LLM USAGE

We used large language models solely for editorial assistance (grammar, phrasing, and clarity). No model was used to generate technical content, derive equations, design experiments, analyze results, or write code. All datasets, algorithms, and empirical results originate from the authors' implementations and public benchmarks. No proprietary or sensitive data were submitted to third-party services.

