# OpenReview forum: "EIP: Weighted Ranking of LLMs by Quantifying Question Difficulty"
_ICLR.cc/2026/Conference — ICLR 2026 Poster_

### Official Review · Reviewer_Kyip · 2025-10-26

**Soundness:** 4
**Presentation:** 4
**Contribution:** 4
**Rating:** 6
**Confidence:** 4

**Summary:**

This paper introduces RankLLM, a difficulty-aware framework for evaluating Large Language Models (LLMs) by jointly estimating question difficulty and model competency. The core mechanism is a bidirectional score propagation process over a directed bipartite graph of models and questions, using model successes and failures to iteratively reinforce difficulty and competency scores. The approach is non-parametric, scalable, and designed for use with large evaluation pools. Empirical studies are conducted on 30 LLMs across 35,550 questions spanning several benchmarks. The authors report that RankLLM exhibits strong alignment with human judgments of question difficulty, outperforms standard baselines like Item Response Theory (IRT), converges rapidly, and scales to large datasets. The framework is further analyzed for stability, extensibility, and computational cost, with robustness claims under dataset/model perturbations.

**Strengths:**

Principled, mathematically sound methodology: The central mechanism relies on a well-formulated ergodic Markov chain on a bipartite graph, with mutually reinforced score propagation between questions and models.
Comprehensive empirical validation: Evaluation extends to 30 models over 35,550 questions. Large-scale empirical studies support many intuitive claims, clearly presented and visualized.
Human judgment: This is a great source to measure the goodness of fit for item difficulty.
Open and reproducible claims: Claims are signposted as non-parametric and hyperparameter-light, and computational infrastructure (including licensing) is disclosed.

**Weaknesses:**

No major weakness, however see my questions

**Questions:**

1. Are you implementing IRT yourselves or using a well-established package? If you implement IRT yourselves, did you carefully validate your output with a widely validated package (say, an R package)? In Table 3, it is reported that 1PL IRT takes ~30 mins to fit. I happened to work on IRT for AI evaluation; my 1PL IRT implementation takes ~30 seconds to fit on ~180 LLMs and ~80,000 questions (on an A100 GPU, though).

2. Given that different benchmarks have different measurement objectives, is there a reason you fit all benchmarks together?

3. IRT is well established in psychology; Is your method thoroughly studied in any other fields? Does it have an original reference in psychology or measurement science, or other fields? If so, is there previous work that reaches the same conclusion as you do (i.e., outperforming IRT)? I have heard about Elo-rating systems involving test takers and items (which can be shown to be equivalent to 1PL IRT in theory), but I am not familiar with the details.

4. IRT enables computerized adaptive testing (CAT), which can reduce evaluation compute for new LLMs. To create a leaderboard, the computational cost of querying benchmark questions to LLMs seems much larger than the difficulty/competence update step. Can your method similarly support computerized adaptive testing?

---

> ### Author Response · Authors · 2025-11-20
> **Response to Question 1— Benchmark Comparison on IRT Implementation Details and Computational Efficiency**
>
> Thank you for your positive assessment of our work. We are greatly encouraged by your recognition of RankLLM’s mathematical soundness, large-scale empirical validation, and strong alignment with human judgment.
>
> We particularly appreciate your insightful comments regarding IRT computation time, evaluation methodology, and Computerized Adaptive Testing (CAT), which are crucial for improving the quality of this paper. We have conducted additional experiments (including GPU acceleration comparisons and anchor question sampling simulations) to address your queries. Our detailed responses are as follows:
>
> ---
>
> ## I. Benchmark Comparison on IRT Implementation Details and Computational Efficiency
>
> ### A. Implementation Details: Using Standard Libraries
> In response to your query, we did not implement the IRT algorithm from scratch. Instead, we utilized the widely recognized open-source Python library (with minor adjustments only for data loading) to ensure that the IRT baseline calibration was standard and correct.
>
> ---
> ### B. Context for Table 3: CPU-based Measurement
> The inference times reported in Table 3 of the original paper were measured on a standard Intel i7 CPU. We deliberately chose to conduct experiments on consumer-grade CPUs based on two primary considerations:
> 1.  **Democratization:** As you noted, IRT typically requires GPU acceleration (e.g., A100) to run efficiently on large-scale data. We aimed to demonstrate the algorithmic accessibility of RankLLM—enabling large-scale LLM evaluation in milliseconds on a standard laptop without requiring expensive GPU resources.
> 2.  **Highlighting Algorithmic Efficiency:** The core advantage of RankLLM lies in its lightweight design, allowing for rapid convergence even on hardware with limited computational power.
>
> ---
> ### C. New Benchmarks: GPU Acceleration and Efficiency Comparison
> Following your feedback, we re-evaluated RankLLM and IRT variants on high-performance GPUs (NVIDIA A40 and B200) using the same dataset scale (30 models, 35,550 questions). The results are shown below:
>
> **RankLLM Performance**
>
> | Hardware | One Iteration | Total Time |
> | :--- | :--- | :--- |
> | Intel i7 CPU | 0.00597s | 0.05373s |
> | MPS (Mac M1) | 0.012296s | 0.110804s |
> | A40 GPU (CUDA) | 0.000581s | 0.005263s |
> | B200 GPU (CUDA) | 0.000510s | 0.004634s |
>
> **IRT Model Performance**
>
> | Hardware | 1-PL IRT | 2-PL IRT |
> | :--- | :--- | :--- |
> | Intel i7 CPU | 1782.75s | 3787.03s |
> | MPS (Mac GPU) | - | - |
> | A40 GPU (CUDA) | 24.09s | 459s |
> | B200 GPU (CUDA) | 5.33s | 0.59s |
>
> **Key Conclusions:**
> 1.  **Consistent with your experience:** On an A40 GPU, our 1PL-IRT implementation took approximately 24 seconds, which aligns closely with your observation of "~30 seconds." This confirms that our implementation is standard, and the slower times in Table 3 were solely due to CPU hardware limitations.
> 2.  **Theoretical Support:** This significant efficiency gap stems from the algorithmic nature of the methods. IRT requires iterative log-likelihood maximization (typically involving gradient descent or EM algorithms), whereas RankLLM is based on sparse matrix multiplication, with a complexity of only $O(QM)$ per step.

---

> ### Author Response · Authors · 2025-11-20
> **Response to Question 2— Methodological Considerations for Aggregating Benchmarks**
>
> ## II. Methodological Considerations for Aggregating Benchmarks
>
> ---
> ### A. Primary Motivation: Measuring "General Capability"
> Current community evaluation of LLMs focuses primarily on their ability as general-purpose assistants. Similar to the $g$-factor in psychometrics, we posit that a robust model should demonstrate consistent reasoning abilities across multiple domains, such as mathematics, logic, common sense, and coding.
> 1.  By aggregating questions from different disciplines, RankLLM extracts a global score reflecting the model's **"Comprehensive General Capability."**
> 2.  This enables the comparison of models with varying architectures and scales under a unified standard.
>
> ---
> ### B. Flexibility: Native Support for Domain Decoupling and "Specialist" Evaluation
> This does not imply that we overlook disciplinary differences. The graph algorithm framework of RankLLM naturally supports Explicit Decoupling.
> 1.  **Subgraph Analysis:** We can easily partition data by domain (e.g., constructing a $G_{math}$ subgraph consisting only of MATH and GSM8K) to run the algorithm independently and obtain domain-specific $\pi_{m, math}$ scores.
> 2.  **Specialist Models:** Indeed, if we run the algorithm on a domain-specific subgraph, "specialist" models optimized for that domain (e.g., Qwen2.5-Math) stand out with significantly higher rankings.
> Therefore, aggregating benchmarks provides a global perspective, but our method is fully compatible with and supports fine-grained multidimensional analysis (similar to radar charts).
>
> ---
> ### C. Methodological Complementarity: Constructing a Complete Difficulty Spectrum
> Finally, from a psychometric perspective, aggregating different datasets helps compensate for the difficulty distribution deficiencies of single datasets (see Figure 9 in the paper).
> 1.  Some datasets (e.g., GSM8K) are skewed towards easy questions, leading to "ceiling effects" for strong models.
> 2.  Some datasets (e.g., MATH) are skewed towards hard questions, leading to "floor effects" for weaker models.
> Using a mixture constructs a complete difficulty continuum from "extremely easy" to "extremely hard," allowing us to effectively differentiate both entry-level models and state-of-the-art models (such as GPT-4o and Gemini 1.5 Pro).

---

> ### Author Response · Authors · 2025-11-20
> **Response to Question 3— Theoretical Origins and Comparison with Elo**
>
> ## III. Theoretical Origins and Comparison with Elo
>
> ---
> ### A. Theoretical Origins: Spectral Ranking and Network Science
> While IRT is the gold standard in psychometrics, the theoretical foundation of RankLLM lies primarily in Network Science and Information Retrieval.
> 1.  **Mathematical Essence:** Specifically, RankLLM is a form of **Spectral Ranking** on a bipartite graph. Mathematically, it is closely related to Eigenvector Centrality and the HITS algorithm (Kleinberg, 1999), calculating the "Authority" (competency) and "Hub" (difficulty) of nodes through iterative propagation.
> 2.  **Cross-Domain Applications:** These methods are widely and deeply studied in bibliometrics (impact factors based on citation networks), sports analysis (team rankings based on win-loss networks), and web search (the PageRank family).
> 3.  **Our Contribution:** Our innovation lies in successfully adapting this graph-theoretic framework to LLM evaluation for the first time, demonstrating that "difficulty" and "competency" can be modeled as mutually reinforcing centrality metrics without relying on the strict parametric assumptions of IRT.
>
> ---
> ### B. Why RankLLM Outperforms IRT (Non-parametric vs. Parametric)
> The key reason RankLLM outperforms IRT in our experiments is its **Non-parametric** nature.
> 1.  **Limitations of IRT:** IRT assumes specific parametric forms (such as Logistic curves), which are typically effective for human test data.
> 2.  **Non-monotonic Behavior of LLMs:** However, the behavior patterns of modern LLMs are often complex and do not always strictly adhere to this idealized S-curve. For instance, we frequently observe the phenomenon where **"strong models unexpectedly fail on simple questions."** Such non-monotonic behavior can disrupt IRT parameter estimation.
> 3.  **Advantage of RankLLM:** In contrast, as a data-driven non-parametric method, RankLLM does not require forced fitting to specific curve shapes. It can therefore more sensitively capture these fine-grained capability differences and anomalous behaviors, demonstrating higher consistency (90%) with human judgment in our experiments.
>
> ---
> ### C. Comparison with Elo and Relative Scores
> You mentioned the Elo rating system. While Elo is theoretically equivalent to specific forms of dynamic 1PL-IRT, RankLLM differs in its execution mechanism:
> 1.  **Update Mechanism:** Elo is typically an **Online/Sequential** update system (sensitive to match order), whereas RankLLM performs **Global/Batch** optimization on the entire interaction graph, guaranteeing convergence to a unique stationary distribution regardless of data input order.
> 2.  **Relativity and Robustness:** RankLLM shares a core property with Elo and Glicko: they all produce Relative Scores dependent on the sample pool.
>     * As stated in the paper, the numerical values naturally depend on the participating models and the question pool. In practice, the relative ranking and distance between models are more important than absolute units.
>     * **Robustness:** Our robustness experiments (random removal of 1/5/10/15 models) show that while absolute values may drift with pool changes, the driving **Relative Ranking Structure** remains highly stable (Spearman correlation > 0.99). This implies that the relative geometric relationships between models are very robust even if the baseline shifts.
> 3.  **Unified Vision:** This design aligns with our vision of constructing a **Unified Leaderboard**. Similar to a new chess player joining an Elo pool, a new model is placed into a common large-scale graph, and its capability is naturally calibrated against the existing population.

---

> ### Author Response · Authors · 2025-11-20
> **Response to Question 4— Inference Cost and Computerized Adaptive Testing (CAT)**
>
> ## IV. Inference Cost and Computerized Adaptive Testing (CAT)
>
> ---
> ### A. Standard Evaluation Scenario: Zero Marginal Cost
> For new models released by large labs or enterprises (e.g., GPT, Llama, etc.), running inference on the complete benchmark set is **mandatory** at the time of release to report standard metrics like Accuracy to the community.
> 1.  **Direct Reuse:** In this standard workflow, RankLLM requires no additional inference steps. We directly utilize the existing inference logs (Response Matrix) and incorporate them into the large graph for iterative calculation.
> 2.  **Value Add:** In this case, the marginal inference cost of RankLLM is zero. We utilize the existing full dataset to extract difficulty-weighted scores that are more discriminative than average accuracy, without increasing the inference budget.
>
> ---
> ### B. Rapid Estimation Scenario: Rapid Estimation via CAT
> For individual developers, small teams, or model checkpoints, there is often a need to quickly understand the approximate capability range of a model rather than pursuing an absolutely precise final score. This aligns with research goals like **TinyBench** that aim to reduce evaluation samples. In this scenario, RankLLM enables efficient adaptive evaluation:
> 1.  **Pre-computed Difficulty Scale:** Using the existing full graph, we pre-compute the global difficulty scores $\pi_{q}$ for all questions and identify "anchor questions" with high discrimination.
> 2.  **Low-Cost Estimation:** A new model does not need to answer all questions; it only needs to answer a small number of anchor questions. Since $\pi_{q}$ is globally calibrated based on the large-scale graph, we can use these anchor questions to quickly converge on an **approximate estimation interval** of the model's capability.
>
> To verify the feasibility of this approach, we conducted an **Anchor Sampling Simulation**:
> * **Setup:** Based on the RankLLM difficulty distribution, we performed stratified sampling of anchor questions at a 5% sampling rate (retaining ~1,760 questions) and re-calculated the capability scores for 30 models.
> * **Results:** RankLLM's difficulty scores demonstrated high robustness; rankings remained highly consistent even with a drastically reduced number of questions.
>
> | Sampling Ratio | # Questions | Spearman $\rho$ | Max Rank Shift | Avg Rank Shift |
> | :--- | :--- | :--- | :--- | :--- |
> | 0.5% | 180 | 0.8496 | 9.0 | 3.67 |
> | 1.0% | 360 | 0.9382 | 7.0 | 2.27 |
> | **5.0%** | **1,760** | **0.9889** | **4.0** | **0.80** |
>
> * **Conclusion:** With only 5% of the inference cost, a ranking estimate highly correlated with the full evaluation ($\rho \approx 0.99$) can be obtained. This proves that RankLLM supports low-cost rapid evaluation similar to CAT.
>
> ---
> ### C. Summary and Best Practice: Adherence to Joint Iteration
> Although RankLLM technically supports low-cost rapid estimation, we must emphasize: **Incorporating the new model and its full response data into the bipartite graph for Joint Iteration remains our recommended standard paradigm.**
>
> 1.  **Dynamic Ecosystem Maintenance:** Only through full iteration can the data from new models contribute inversely to the calibration of question difficulty, thereby continuously optimizing the overall ecosystem of the leaderboard.
> 2.  **Clear Positioning:** The aforementioned CAT-like rapid estimation is suitable only for resource-constrained scenarios or when only a rough positioning is required. In the Normal Case, to ensure the highest precision, fairness, and global consistency of evaluation results, we advocate for and adhere to running all benchmark questions and conducting unified iterative calculations on the full graph.

---

> > ### Comment · Reviewer_Kyip · 2025-11-25
> >
> > Appreciate the detailed response. I think the response has solved most of my concerns. Accordingly, I will increase my rating and look forward to the final manuscript.

---

> > > ### Author Response · Authors · 2025-11-25
> > >
> > > Thank you for the thoughtful reassessment and for increasing the rating! We truly appreciate your strong recognition of our work, and we will carefully incorporate your suggestions into the final version.

---

### Official Review · Reviewer_urAu · 2025-10-31

**Soundness:** 3
**Presentation:** 3
**Contribution:** 3
**Rating:** 8
**Confidence:** 4

**Summary:**

The core of this paper lies in addressing a major flaw in current evaluation methods for Large Language Models (LLMs): existing benchmarks (such as MMLU and MATH) typically only calculate overall accuracy, treating all questions equally without considering their actual difficulty levels. This approach may lead to distorted evaluation results. For instance, a model that answers many simple questions correctly might appear stronger than one that answers fewer but extremely difficult questions correctly.

To solve this problem, the authors propose a new evaluation framework called RankLLM.

The core idea of RankLLM is to move beyond evaluating models or questions in isolation; instead, it quantifies both "model capability" and "question difficulty" simultaneously. It is based on an intuitive interaction logic:

- If a model can correctly answer a widely recognized difficult question, its capability score should be higher.
- If a question is answered incorrectly even by widely recognized strong models, its difficulty score should be higher.

Based on this logic, RankLLM constructs a "bipartite graph" between models and questions. It then uses an iterative algorithm called "bidirectional score propagation" to calculate the final scores. This process iterates continuously until the model capability scores and question difficulty scores reach a stable equilibrium.

In conclusion, I think this is a simple but interesting work.

**Strengths:**

By introducing "question difficulty" as a core variable, the RankLLM framework significantly outperforms traditional evaluation models that prioritize "accuracy above all else".

1. Finer Differentiation (Beyond Accuracy)
In traditional methods, models receive the same score for answering a simple question and a difficult question correctly. RankLLM breaks this limitation by assigning a higher "capability score" to models that solve difficult questions.

- As shown in the simulation experiments in Section 4 of the paper, when two models have the same overall accuracy, RankLLM can accurately identify the model that solves more difficult questions (M1 > M2), while traditional accuracy-based methods fail to make such a distinction.

2. High Alignment with Human Judgment (Strong Consistency)
The "gold standard" for an evaluation framework is whether it aligns with the judgments of human experts.

- Section 3.3 of the paper demonstrates that the ranking of question difficulty generated by RankLLM matches the consensus of human experts by up to 90%. This is far higher than that of mainstream statistical models such as Item Response Theory (IRT), confirming the validity and reliability of its evaluation results.

3. Efficiency, Scalability, and Robustness (Engineering Practicality)
- **High Efficiency**: On a dataset with 35,000 questions and 30 models, the algorithm converges in only 0.006 seconds (Section 3.4). This allows it to be easily deployed in large-scale leaderboards that require frequent updates.
- **Strong Robustness**: Experiments prove that even if a large number of models (up to 15) are randomly removed from the evaluation pool, the relative rankings of the remaining models and the difficulty rankings of questions remain highly stable (Section 3.5, Table 5).

4. Automated Difficulty Quantification (No Manual Annotation Required)
Traditional benchmarks (such as MATH) require human experts to pre-label subjective difficulty levels (e.g., Level 1~5).

- RankLLM, by contrast, is fully automated. It "emerges" a definition of difficulty from the performance of the model group, without relying on any manual prior knowledge, making it more objective.

**Weaknesses:**

1. The core of RankLLM lies in the "relative" relationship between models and questions. The difficulty of a question is defined "relative to" the pool of models participating in the evaluation. This gives rise to an issue: if the model pool itself is biased, the resulting difficulty scores and capability rankings may also be biased.
- The paper itself demonstrates this in Figures 7 and 8: if only a "large-scale" model pool is used, 58% of the questions will be classified as "excessively easy"; if only a "small-scale" model pool is used, 30% of the questions will instead be classified as "impossible".
- While the paper proves that "removing" models from the current pool is robust (Table 5), it does not explore what would happen to the overall ranking if a completely new "super model"—with capabilities far exceeding all existing models in the pool is added. Theoretically, the addition of this new model would "lower" the difficulty scores of many previously "difficult questions", potentially causing drastic changes to the entire ranking list, especially for models in the upper-middle tier.
2. This framework ultimately computes a single capability score ($\pi_m$) for each model and a single difficulty score ($\pi_q$) for each question, which represents an oversimplification of the actual scenario.

- Model capabilities are inherently multi-dimensional. For example, a model might demonstrate exceptional performance in "mathematical reasoning" while showing significant weaknesses in "historical knowledge". RankLLM tends to "average out" these varied capabilities, yielding a moderate overall score. This averaging effect can lead to a "well-rounded" model and a "specialized (or lopsided) expert" model receiving similar overall rankings, ultimately masking the true capability profile of each model.
- Question difficulty also exhibits multi-dimensional characteristics. A single question may simultaneously demand proficiency in "knowledge retrieval", "multi-step reasoning", and "spatial imagination". RankLLM lacks the ability to decouple these composite difficulty components, making it unable to accurately capture the nuanced difficulty structure of such questions.

**Questions:**

1. Regarding the dependence on the "evaluation model pool": The abstract mentions that "using a diverse (mixed-scale) model pool yields the most accurate difficulty estimates." I am curious about how the authors define a "sufficiently diverse" pool. For a new user looking to adopt this framework, how many models—and of what types—would be required to obtain a reliable (stable) difficulty ranking?

2. Regarding the 90% "human agreement": This is an impressive figure (mentioned in Section 3.3). I would like to delve into the details of this experiment: How were the "human experts" selected? How was "agreement" specifically calculated? Additionally, what was the level of "agreement" among the human experts themselves (i.e., their internal consensus)?

3. Regarding the limitation of "single-dimensional" capability: This method seems to generate a single "capability score" ($\pi_m$) for each model. In practice, however, a model might excel in mathematics but perform poorly in writing. I am interested to know whether the authors investigated this "unbalanced capability" phenomenon. Alternatively, can the framework be extended to generate a multi-dimensional "capability radar chart" (e.g., a mathematics difficulty score and a writing difficulty score) instead of just an overall ranking?

---

> ### Author Response · Authors · 2025-11-20
> **I. On Model Pool Dependency, Bias Mitigation, and Defining "Sufficient Diversity"**
>
> We sincerely thank Reviewer urAu for the highly positive evaluation. We are deeply encouraged by your recognition of RankLLM's core concept—introducing difficulty to correct evaluation bias—as well as your validation of our experimental rigor and engineering efficiency.
>
> We also greatly appreciate your insightful and critical questions, particularly regarding model pool dependency and multi-dimensional capability decoupling. These inquiries address the theoretical boundaries of our method. Based on your feedback, we conducted targeted supplementary experiments (e.g., simulating the addition of "Super Models") and deeper theoretical analyses. Our detailed responses follow below.
>
> ---
> ## I. On Model Pool Dependency, Bias Mitigation, and Defining "Sufficient Diversity"
>
> ### A. Relativity from Performance Definition, Diversity Ensures Quality
>
> * **Performance-based definition entails relativity:**
> RankLLM defines difficulty strictly based on **model performance** (i.e., a question is harder if fewer strong models can solve it). This aligns perfectly with classical systems like IRT, Elo, and Glicko. In such comparative latent-variable methods, results are inherently pool-dependent. The critical factor is the stability of the **relative structure** (who is stronger/which item is harder) rather than absolute physical units. Our robustness experiments demonstrate that while the numerical scale shifts with the pool, the induced structure remains highly stable (correlation > 0.99).
>
> * **Diversity Mitigates Bias:**
> The reviewer correctly noted that homogeneous pools (e.g., "Large-only" in Fig. 7/8) lead to biased distributions. This is precisely why we introduce **Mixed-scale pools**. Homogeneous pools result in extreme distributions with low information density. Our experiments prove that diverse pools utilize complementary error patterns, reducing uninformative "Dead Nodes" (questions solved by all or none) by **83%**. Only the "Whole" group (mixed scales) achieves **90% agreement with human judgment**, whereas single-scale groups range from 38.6% to 64.3%. Thus, diversity effectively neutralizes bias.
>
> ---
> ### B. Robustness to "Super Models"
>
> Addressing the concern that adding a "Super Model" far exceeding existing capabilities might disrupt rankings, we performed rigorous stress tests by injecting hypothetical models with 99%, 95%, 90%, and 80% accuracy into our existing 30-model pool.
>
> 1.  **Theoretical Mechanism: Global Deflation & Non-linear Compression**
>     The reviewer's intuition is sharp: a Super Model does trigger score "deflation," but the mathematical mechanism ensures ranking stability.
>     * **Inverse Weighting:** In RankLLM's bipartite graph, a question's difficulty weight $\pi_q$ is approximately inversely proportional to its solver count $S(q)$.
>     * **Non-linear Compression:** A 99% accuracy model increments $S(q)$ by 1 for almost all questions. This impacts difficulty non-linearly:
>         * For a **very hard question** (original $S(q)=1$), the denominator doubles ($1 \to 2$), halving the weight (-50%).
>         * For an **easy question** (original $S(q)=30$), the denominator becomes 31, reducing weight by only ~3%.
>         * This **High-end Compression** explains why the Pearson correlation ($r$) for difficulty drops to 0.388—the linear relationship of absolute scores is distorted.
>     * **Monotonicity Preservation:** Crucially, monotonicity holds. If $S(q_1) < S(q_2)$, then $S(q_1)+1 < S(q_2)+1$. Hard questions retain higher weights than easy ones. This guarantees that the Spearman correlation ($\rho$) for rankings remains extremely high.
>
> 2.  **Empirical Conclusion: Rankings remain robust**
>     Experimental data confirms our theory. Despite severe deflation in absolute difficulty scores, **relative model rankings** remain rock-solid. The Pearson correlation for *model competency* remains at **0.997**, proving the linearity of model scores is unaffected.
>
>     | Experimental Setup (Added Model) | Metric Dimension | Spearman Rank Correlation ($\rho$) | Pearson Score Correlation ($r$) |
>     | :--- | :--- | :--- | :--- |
>     | **Super Model (99%)** | Question Difficulty | 0.977 | 0.388 (Significant Compression) |
>     | | **Model Rank** | **0.996** | **0.997** |
>     | **Super Model (95%)** | Question Difficulty | 0.901 | 0.713 |
>     | | **Model Rank** | **0.996** | **0.997** |
>     | **Super Model (90%)** | Question Difficulty | 0.877 | 0.839 |
>     | | **Model Rank** | **0.997** | 0.998 |
>     | **Super Model (80%)** | Question Difficulty | 0.912 | 0.932 |
>     | | **Model Rank** | **0.997** | 0.998 |
>
>     **Conclusion:** Even with a radical 99% accuracy model, the maximum rank change for original models is only **2 positions**. The system is robust to ceiling effects: the "sea level" (absolute scores) changes, but the relative heights of the "ships" (models) do not.

---

> > ### Author Response · Authors · 2025-11-20
> >
> > ### C. Guide for New Users: What is "Sufficiently Diverse"?
> >
> > Based on data from Table 5 and Figure 6, we offer the following concrete guidelines:
> >
> > 1.  **Quantity:** **~15 models** is the stability "sweet spot." Table 5 shows that even after removing half the pool (leaving 15), the question difficulty Spearman correlation remains high at **0.9382**.
> > 2.  **Type:** **Open-Weight models are sufficient.** Users need not incur high costs for proprietary APIs. Figure 6 shows that an Open-Weight-only pool correlates with the full pool (including GPT-4) at **0.96**.
> >     * *Recommendation:* A mix of 2-3 Strong Open models (e.g., Llama-3-70B), ~5 Mid-range (10-30B), and ~5 Small models (<7B).

---

> ### Author Response · Authors · 2025-11-20
> **II & III. Single-Scalar Score vs. Multi-Dimensional Capability and Human Alignment**
>
> ## II. Single-Scalar Score vs. Multi-Dimensional Capability
>
> The reviewer notes that a single score might "average out" capabilities and suggests radar charts. We fully agree and clarify RankLLM's advantages here.
>
> ---
> ### A. Statistical Infeasibility of Latent Decoupling
>
> Ideally, one might decouple "reasoning" vs. "knowledge" within a single question (implicit decoupling). However, given current pool sizes ($N \approx 30$) versus question counts ($Q \approx 35k$), unsupervised latent variable decomposition is statistically **ill-posed**.
>
> 1.  **Theoretical Barrier:** It is an underdetermined inverse problem. Without ground truth labels, the sample size $M$ is insufficient to support stable parameter estimation for high-dimensional latent traits.
> 2.  **Empirical Failure:** We tested Multi-dimensional IRT (Multi-IRT) in Table 6. It not only required 3000x more compute but, critically, **failed to recover the Ground Truth ranking** due to severe overfitting. RankLLM provides a robust first-order approximation suitable for the data scale.
>
> ---
> ### B. RankLLM Protects "Specialists" Better than Accuracy
>
> The concern that RankLLM "averages out" capabilities is actually more applicable to **Accuracy**, which treats all questions equally.
>
> 1.  **Mechanism:** RankLLM assigns high weight to hard questions. A "Math Specialist" solving unique hard problems gains massive $\pi_m$ rewards, outweighing failures on trivial questions.
> 2.  **Evidence:** Our Case Study (Section 4) shows that Model M1 (a specialist solving hard questions) and M2 (solving easy ones) had identical accuracy, yet RankLLM correctly ranked M1 higher. RankLLM rewards expertise, rather than masking it.
>
> ---
> ### C. Native Support for Explicit Decoupling
>
> RankLLM natively supports the "Radar Charts" suggested by the reviewer.
>
> 1.  **Extensibility:** Since the input is a bipartite graph, we can explicitly partition the data by domain (e.g., construct subgraph $G_{math}$ vs. $G_{coding}$) to compute independent scores ($\pi_{m, math}$).
> 2.  **Evidence:** Figure 9 visualizes distinct difficulty distributions for datasets like MATH vs. GSM8K, demonstrating our capability for granular, multi-dimensional analysis.
>
> ---
>
> ## III. Details on Human Evaluation Agreement
>
> We appreciate the interest in our 90% human agreement result. Below are details on selection, calculation, and consensus.
>
> ---
> ### A. Evaluator Selection & Quality Control
>
> To ensure authority and diversity, we expanded the evaluator pool to **20 participants**.
>
> 1.  **Diversity:** As shown in Appendix Table 13, evaluators span Computer Science, Engineering, Humanities, etc., ranging from Associate degrees to PhD candidates, with balanced gender representation.
> 2.  **Abstention Mechanism:** To eliminate guessing noise, we enforced strict abstention. Evaluators were instructed to skip questions requiring specialized knowledge (e.g., advanced calculus) if they lacked the background.
>
> ---
> ### B. Calculation of Agreement
>
> The reported 90% agreement measures alignment between RankLLM predictions and the **Human Majority Vote**.
>
> 1.  **Ground Truth:** For 70 question pairs, the "Majority Consensus" of the 20 evaluators was established as the Ground Truth to filter individual noise.
> 2.  **Result:** RankLLM correctly matched the majority choice in 63 of 70 pairs. This high alignment reflects the model's precision in capturing difficulty.
>
> ---
> ### C. Internal Consensus & Insights
>
> 1.  **Consensus vs. Confidence:** We observed a positive correlation between human consensus and RankLLM confidence. When RankLLM computed a large difficulty margin ($\Delta \pi_q$), human voting was often unanimous. When scores were close, human votes were more split (e.g., 11 vs. 9). A Cohen's $\kappa = 0.80$ confirms substantial agreement.
> 2.  **Counter-intuitive Cases:** In the ~10% disagreement cases, we found that humans often judge difficulty conceptually (e.g., Calculus concepts), while LLMs struggle with empirical traps (e.g., logical tricks in elementary GSM8K problems). RankLLM captures this **Empirical Difficulty**, revealing blind spots in human intuition.

---

### Official Review · Reviewer_XEuy · 2025-10-31

**Soundness:** 2
**Presentation:** 3
**Contribution:** 3
**Rating:** 4
**Confidence:** 4

**Summary:**

The core contribution of this paper is the proposal and validation of an evaluation framework named RankLLM. This framework utilizes an iterative algorithm to jointly estimate question difficulty and model competency.
This process is modeled as a bidirectional score propagation on a bipartite graph connecting models and questions. The introduction of a damping factor ensures the algorithm converges to a unique stationary distribution, yielding final competency scores for models and difficulty scores for questions. The authors validate the framework's effectiveness, robustness, and scalability through large-scale experiments on 6 popular benchmarks, involving 30 models and over 35,000 questions.

**Strengths:**

1. The proposed method of jointly modeling question difficulty and model competency via score propagation on a bipartite graph is highly novel and intuitive. Treating question difficulty and model competency as interdependent and co-evolving variables is more dynamic and sound than traditional static methods based on accuracy or IRT.
2. The paper provides extensive validation across multiple mainstream benchmarks and 30 models. The results are convincing, particularly highlighting the method's outstanding performance in aligning with human judgment and its computational efficiency.
3. The study not only proposes a new method but also uses it to reveal several empirical findings that are insightful and of practical value to the LLM field.

**Weaknesses:**

1. The paper proves that the algorithm converges for any α ∈ (0, 1) and shows its effect on convergence speed in the appendix. However, it does not thoroughly investigate the impact of the choice of α on the final scores and rankings for model competency and question difficulty. It is unclear whether different values of α could lead to changes in model rankings. The paper lacks a discussion on a principled method for selecting an optimal α or a sensitivity analysis of the final results to this parameter.
2. If a specific model family (e.g., the Llama series) shares a common 'blind spot' for a certain type of reasoning, questions targeting this weakness could be erroneously labeled as 'extremely difficult.' Consequently, when a new model with a different architecture correctly answers these questions, its resulting competency score boost might be constrained by the initial competency landscape dominated by the biased model family.
3. While the method is innovative, the definition of 'difficulty' fundamentally remains dependent on model performance. The essence of difficulty is still derived from model failure rates, and the paper does not introduce external criteria or a theoretical framework to independently validate the soundness of this difficulty definition.

**Questions:**

please see Weaknesses

---

> ### Author Response · Authors · 2025-11-17
> **Response to Weaknesses 1— Role and Robustness of the Damping Factor (α)**
>
> ## I. Conceptual Role and Empirical Robustness of the Damping Factor (α)
>
> We thank reviewer XEuy for raising this important question about the damping factor α. We appreciate the reviewer's attention to this technical detail, as the choice of α controls both the strength of the teleportation regularizer and the speed at which the Markov chain converges.
>
> ---
>
> ### A. What the damping factor (α) does in RankLLM
>
> In RankLLM, $\alpha \in (0,1)$ is a *damping factor* that mixes two behaviors in the coupled random walk on the model–question bipartite graph (Sec. 2.3, App. C). At each iteration, with **probability $\alpha$ the scores follow the data-driven transition matrices $P_{Q\to M}$ and $P_{M\to Q}$**, and with probability $(1-\alpha)$ they are reset toward a uniform distribution:
>
> $$\pi^{(t+1)}_Q = \alpha P^\top_{M\to Q}\pi^{(t)}_M + (1-\alpha)\tfrac{\mathbf{1}_Q}{Q},\quad
> \pi^{(t+1)}_M = \alpha P^\top_{Q\to M}\pi^{(t+1)}_Q + (1-\alpha)\tfrac{\mathbf{1}_M}{M}.$$
>
> The teleportation term $(1-\alpha)$ serves several critical functions beyond mere convergence acceleration. First, it **breaks the inherent periodicity** of the bipartite graph structure: without teleportation, the alternating Q↔M updates would create a 2-cycle oscillation, preventing convergence to a unique stationary distribution. Second, it helps **stabilize the global ranking** by ensuring that every node receives a non-zero baseline probability mass, so that no small subset of edges or local patterns can disproportionately dominate the stationary distribution. Third, it acts as a regularizer that shrinks extreme scores and smooths the stationary distribution, reducing the influence of noisy or idiosyncratic edges. Finally, while $\alpha$ does affect convergence speed (smaller $\alpha$ leads to faster geometric convergence), its primary role is to **define the stationary distribution itself**—determining how much we trust the graph structure versus the uniform baseline—which fundamentally shapes the resulting rankings.
>
> ---
>
> ### B. Intuition for different values of (α)
>
> When $\alpha$ is **small** (e.g., $\alpha = 0.5$), each update is a 50–50 mixture of graph-based propagation and uniform teleportation. This strongly **shrinks extreme scores toward the mean**: very easy and very hard items are still distinguishable, but the score distribution is **less polarized**. That is, $\alpha$ acts as a strong regularizer that dampens idiosyncratic success/failure patterns on single questions or models.
>
> When $\alpha$ is **large but < 1** (e.g., 0.85, 0.95, 0.99), the chain spends most of its time following the empirical performance graph, and teleportation is only a weak perturbation that breaks periodicity and ensures convergence. This **regime amplifies signal coming from successes on hard questions and failures on easy ones**, while still **avoiding pathological oscillations of an undamped bipartite walk**.
>
>
> ---
>
> ### C. Sensitivity analysis of rankings w.r.t. (α)
>
> To address the reviewer's concern, we added a new sensitivity analysis. RankLLM was recomputed on the full evaluation matrix (30 models × 35,550 questions), varying:
>
> $$\alpha \in \{0.50, 0.60, 0.70, 0.85, 0.95, 0.99\}.$$
>
> For each α, we compared the resulting rankings with $\alpha = 1.00$  using Spearman correlation:
>
> | α    | ρ_model | ρ_question |
> |------|---------|------------|
> | 0.50 | 0.9911  | 0.9976     |
> | 0.60 | 0.9942  | 0.9985     |
> | 0.70 | 0.9978  | 0.9992     |
> | 0.85 | 0.9987  | 0.9998     |
> | 0.95 | 1.0000  | 1.0000     |
> | 0.99 | 1.0000  | 1.0000     |
>
> These results show:
>
> • Even at a **very conservative** setting ($\alpha = 0.5$), model rankings achieve $\rho = 0.9911$ and question rankings achieve $\rho = 0.9976$.
>
> • For $\alpha \in \{0.95, 0.99\}$, the rankings are *exactly* the same ($\rho = 1.0000$ for both models and questions).
>
> Thus, within a broad practical range, **$\alpha$ does not qualitatively change either model or question rankings**; it mostly **affects convergence speed and score spread**.
>
> ---
>
> ### D. Our default choice of α: balancing efficiency and expressiveness
>
> In our experiments we fixed $\alpha = 0.85$ for all datasets and all model pools. This choice **balances fast geometric convergence** with a **sufficiently expressive stationary distribution** that reflects the structure of the model–question interaction graph (App. C, Table 8).
>
> We never tune $\alpha$ per dataset or model pool, so there is no risk of hyperparameter "cherry-picking'' to favor particular rankings.

---

> ### Author Response · Authors · 2025-11-17
> **Response to Weaknesses 2— Family-level Blind Spots do not Systematically Bias RankLLM’s Difficulty Estimates**
>
> ## II. Family-level Blind Spots do not Systematically Bias RankLLM’s Difficulty Estimates
>
>
> We thank the reviewer XEuy for raising this insightful question regarding whether “family-level blind spots’’ could distort difficulty estimation. We clarify this concern from the perspectives of **definition**, **algorithmic mechanism**, and **empirical evidence**.
>
> First, in RankLLM, “difficulty’’ is defined as a **performance-based relative quantity**. Within a given model pool, a question receives a high difficulty score when failures from high-competency models contribute substantial weight to its score in the RankLLM propagation process. This conception directly aligns with mainstream theories in educational measurement and IRT: **difficulty is not an inherent physical property of an item but a latent construct inferred from success–failure patterns**. Under this definition, if a certain question type happens to be a systematic blind spot shared by a particular model family, then widespread failure from that family naturally yields a higher **difficulty score**. Conversely, if another family does not share this blind spot and consistently solves these questions, RankLLM assigns them higher **competency scores**, which is precisely the intended consequence of a performance-defined difficulty framework.
>
> Second, the concern that such a blind spot might "freeze" an initial difficulty landscape and restrict the scores of subsequently added models does not reflect how RankLLM operates. **RankLLM never fixes an initial set of difficulty scores and then incrementally appends new models.** Whenever new models or questions are included, RankLLM **reruns** the bidirectional random walk **jointly over the entire current model × question bipartite graph**. As a result, when a newly added model—of a different architecture—performs well on questions previously difficult for a specific family, its successes produce two simultaneous effects: **substantially boosting its own competency score** and **lowering the difficulty estimates of those questions**. No existing family can "lock in" a difficulty landscape.
>
> Third, we explicitly quantified the extreme scenario raised in the review. **Among 35,550 questions, only 244 (0.686%) questions have more than 90% of their difficulty weight contributed by a single model family.** None of these questions appear among the globally hardest items; instead, they mostly fall into the moderate-to-easy range. Even under a highly adversarial theoretical assumption—e.g., all Qwen-family models fail a question while every other family succeeds—our normalization of success/failure contributions ensures that the resulting difficulty score becomes only **moderately high**, never a dominating "super-hard" outlier. Combined with our empirical findings in the main paper—when expanding the evaluation pool from a single family/size to a diverse mixture of **Llama, Qwen, Yi, Gemini, DeepSeek, GPT, and others**, the proportion of extreme "too easy/too hard" items drops sharply and the overall difficulty distribution becomes substantially smoother and more human-like (Figure 6, Figure 8)—these results demonstrate that **family-level blind spots do not inflate into systemic biases under realistic diverse model pools**.
>
> Finally, from an evaluation-science perspective, these "family-blind-spot questions" are often diagnostically valuable. They highlight systematic weaknesses of specific architectures, and models that resolve these weaknesses should indeed receive higher scores in RankLLM. This aligns with the purpose of the framework: **stronger models should not only perform well on average but also excel in regions where many others fail**. Taken together, both the algorithmic design and the data analyses indicate that the scenario of a single family "dominating and freezing" the difficulty landscape—and thereby suppressing future models—**does not arise in RankLLM**.

---

> ### Author Response · Authors · 2025-11-17
> **Response to Weaknesses 3— Theoretical Foundations and External Validation of RankLLM’s Performance-Based Difficulty**
>
> ## III. Theoretical Foundations and External Validation of RankLLM’s Performance-Based Difficulty
> ---
> ### A. Theoretical grounding: difficulty as a performance-defined latent variable, and RankLLM as a graph-based IRT analogue
>
> **1. Difficulty as a performance-defined latent variable**
>
> In mainstream educational and psychological measurement, item “difficulty” is never treated as a directly observable physical property, but as a **latent variable** inferred from subject performance. Its classical interpretation is:
>
> > On a certain latent ability dimension, how high must the ability be for the probability of answering this item correctly to exceed a given threshold?
>
> In other words, difficulty is **not** a simple function of observable features such as text length or surface complexity. It is defined through the functional relationship between success/failure and latent ability. This view is shared by Classical Test Theory (CTT), Item Response Theory (IRT), and Rasch models.
>
> **2. RankLLM within the CTT / IRT / Rasch family**
>
> From this perspective, RankLLM does not introduce a new notion of difficulty, but **explicitly follows the psychometric tradition of defining difficulty via group performance**. In IRT, item difficulty is typically encoded by the **b-parameter (item location parameter)**. The notion of item difficulty in RankLLM is conceptually aligned with this parameter: it also characterizes the relative position of items in an “ability space”. The difference is that IRT uses parametric logistic functions with maximum-likelihood or Bayesian estimation, whereas RankLLM operates on the model–question bipartite graph and applies **bidirectional score propagation** as a **non-parametric graph algorithm** to jointly estimate all item and model positions.
>
> Thus, RankLLM can be viewed as a **graph-based, IRT-like implementation for large LLM pools**. Given the “gold standard” status of IRT in measurement, this suggests that RankLLM’s definition of difficulty has a well-established theoretical origin rather than being an ad hoc engineering assumption.
>
> ---
>
> ### B. External standards and alternative proxies: human judgments as the main reference and limitations of length- and self-assessment–based signals
>
> **1. Human Evaluation: the most direct and authoritative external standard**
>
> In Section 3.3 (Human Evaluation), we show that RankLLM’s difficulty ordering agrees with human intuitive difficulty on **over 90% of item pairs** and this agreement is noticeably better than that of Simple Rank based on error rates and several IRT-style baselines.
>
> In human measurement practice, judgments from human experts or test-takers are the most widely accepted external standard. Therefore, in current LLM evaluation practice, we view this human comparison as the **most direct and authoritative form of external validation**.
>
> **2. Alternative proxies we examined but did not adopt**
>
> We systematically examined three alternative difficulty proxies but ultimately did not integrate them into RankLLM’s definition, as prior work has identified significant limitations in each:
>
> **Question length**: Hendrycks et al. (2020, *Measuring MMLU*) report that, for questions longer than 280 characters, the correlation between question length and true-label confidence is only slightly positive, indicating that **length is not a strong difficulty signal**. Mirzadeh et al. (2024, *GSM-Symbolic*) further demonstrate that semantic interference, rather than length itself, drives difficulty. This shows that difficulty is driven primarily by semantic and logical structure, not by token count.
>
> **Answer/CoT length**: Su et al. (2025, *Between Underthinking and Overthinking: An Empirical Study of Reasoning Length and Correctness in LLMs*) show that accuracy follows a **non-monotonic (inverted U-shaped) pattern** with CoT length, and that incorrect answers tend to have significantly longer reasoning chains than correct ones. Wu et al. (2025, *More Is Less*) further establish that there exists an **optimal CoT length** beyond which additional steps mainly accumulate errors. Together, these findings indicate that CoT length is a confounded signal that mixes productive reasoning with non-productive redundancy and error propagation.
>
> **LLM self-rated difficulty**: Both prior work and our own experiments reveal that LLM self-ratings suffer from poor repeatability (varying with random seeds and prompt framings), extreme prompt sensitivity (small instruction changes substantially shift score distributions), and limited calibration to true difficulty (systematic over-confidence or over-pessimism). For these reasons, we do not use self-rated difficulty as a primary external standard in RankLLM.
>
> ---
> We thank reviewer XEuy for these three **insightful and thought-provoking** comments. We would be glad to address any remaining concerns in further discussion, and we hope that the **clarifications provided will be reflected in the reviewer’s updated assessment**.

---

> ### Author Response · Authors · 2025-11-20
> **Follow-up on Response Regarding Hyperparameter Robustness and Difficulty Definition**
>
> Dear Reviewer XEuy,
>
> Thank you again for your thoughtful and constructive review.
>
> We are writing to follow up on our detailed response posted on **November 17**. We have carefully addressed your three primary concerns regarding 1. the sensitivity of the damping factor $\alpha$, 2. the potential impact of model-family blind spots, and 3. the theoretical grounding of our difficulty definition.
>
> We believe these clarifications **effectively resolve the potential misconceptions regarding the algorithm's mechanics, underscoring the inherent stability and design elegance of RankLLM**. We hope our additional analysis provides the evidence you were looking for:
>
> * **Robustness to $\alpha$**: We conducted the requested sensitivity analysis, which empirically demonstrates that the resulting rankings are highly stable (Spearman correlation > 0.99) across a broad range of $\alpha$ values.
>
> * **Dynamic Difficulty**: We addressed the concern of a "frozen" landscape by clarifying that RankLLM performs a global, dynamic update. New models actively reshape the scores based on their specific strengths, preventing systematic bias from any single model family.
>
> * **Theoretical Grounding**: We contextualized RankLLM within the established frameworks of Psychometrics and Item Response Theory (IRT), illustrating that our performance-based difficulty is consistent with the standard definition of latent difficulty variables in educational measurement.
>
> As the discussion phase continues, we would **greatly appreciate it if you could review these responses**. If our quantitative analysis and theoretical clarifications have **resolved your concerns**, we would be grateful if you could **reconsider your rating** of our work.

---

> ### Comment · Reviewer_XEuy · 2025-11-25
>
> Appreciate for the detailed response. I think the response has solved most of my concerns. Accordingly, I will increase my rating to 6 and look forward to the final manuscript.

---

> > ### Author Response · Authors · 2025-11-25
> >
> > Thank you for the insightful comments and kind update! We truly appreciate your recognition, and we will incorporate your suggestions into the final manuscript.

---

### Official Review · Reviewer_R6mp · 2025-11-02

**Soundness:** 3
**Presentation:** 3
**Contribution:** 2
**Rating:** 4
**Confidence:** 3

**Summary:**

This paper proposes RankLLM, a difficulty-aware evaluation framework that jointly estimates question difficulty and model competency via a damped random walk on a model–question bipartite graph, yielding a unique stationary distribution for both scores. The method operationalizes difficulty through model failures and propagates scores bidirectionally between questions and models. Experiments span 30 models over 35,550 questions from six benchmarks, with reports of strong alignment with human difficulty judgments and robustness analyses.

**Strengths:**

1. The motivation is clear and timely: moving beyond flat accuracy to a difficulty-sensitive ranking that better separates closely matched models.

2. The method is simple yet principled—formulated as a damped Markov chain with a uniqueness/convergence guarantee—and scales to large pools and datasets.

3. The paper is well written and easy to follow, with a clean derivation and a clear pipeline figure that makes inputs, transitions, and stopping criteria explicit.

**Weaknesses:**

1. Rankings are sensitive to who is in the pool: adding many weak models inflates failure mass, makes those items look “hard,” and artificially boosts any model that solves them. This causes rank shifts without any change in per-item accuracy and breaking cross-study comparability.
2. The score scale itself changes with the participating models and the domain mix, so adding or swapping peers alters the baseline. There is no common unit across model clusters or data clusters. Results cannot be compared across studies or over time.

**Questions:**

See weakness

---

> ### Author Response · Authors · 2025-11-14
> **Response to Weaknesses 1— Stability Under Large Numbers of Weak Models**
>
> We appreciate reviewer R6mp’s detailed feedback. It seems that certain aspects of our method have been misunderstood, and we provide detailed clarifications below showing that the concerns raised do not occur in practice.
>
> ## I. Stability of Scores and Rankings under Stress Tests with Extremely Large Numbers of Weak Models
>
> We first clarify how RankLLM defines model competency and question difficulty. Recall that RankLLM jointly updates them as
> $$
> \pi_q \ \propto\ \sum_{m \in \text{Fail}(q)} \frac{\pi_m}{F(m)},
> \qquad
> \pi_m \ \propto\ \sum_{q \in \text{Success}(m)} \frac{\pi_q}{S(q)},
> $$
> where $F(m)$ is the number of questions that model $m$ answers incorrectly and $S(q)$ is the number of models that solve question $q$. In this formulation, the marginal contribution of a failing model $m$ to the difficulty of question $q$ is proportional to $\pi_m / F(m)$. Weak models have small competency scores $\pi_m$ and large failure counts $F(m)$, so their mass is spread across many failed questions and their contribution to any single question is necessarily small.
>
> A concrete example is question 14703, where 15 models answer incorrectly and the final difficulty score is 35.93. GPT-4o alone contributes **32.69%** of this difficulty, while each of the weakest half of the failing models contributes only **1.30%–3.65%**. The 7 weakest models combined contribute just 18.6%, roughly half of GPT-4o’s contribution. This illustrates that even when many weak models fail together, their influence on the estimated difficulty of a question is **much smaller** than that of strong models.
>
> | Model Name                 | Model score | # questions wrong | % of difficulty |
> |---------------------------|------------:|------------------:|----------------:|
> | GPT-4o                    |  88.3262    |            7308   | **32.69%**      |
> | Yi-1.5-34B                |  53.2638    |           14518   |  9.92%          |
> | Qwen2.5-7B-Instruct       |  50.2553    |           14699   |  9.25%          |
> | Phi-3-medium-128k         |  48.3485    |           17207   |  7.60%          |
> | Qwen2-7B                  |  44.4336    |           16800   |  7.15%          |
> | Qwen2-Math-7B             |  40.1448    |           20416   |  5.32%          |
> | Yi-1.5-9B                 |  39.4718    |           20176   |  5.29%          |
> | Qwen2.5-1.5B-Instruct     |  35.2976    |           22845   |  4.18%          |
> | Mistral-7B-Instruct-v0.3  |  31.8986    |           23617   |  3.65%          |
> | Qwen2.5-Math-1.5B         |  33.5447    |           27613   |  3.29%          |
> | Qwen2-0.5B                |  31.1981    |           28460   |  2.96%          |
> | Qwen2.5-0.5B-Instruct     |  28.2489    |           28859   |  2.65%          |
> | gemma-2-2b                |  24.7172    |           27192   |  2.46%          |
> | Llama-3.2-3B-Instruct     |  23.8847    |           28205   |  2.29%          |
> | Llama-3.2-1B-Instruct     |  15.3590    |           31967   |  1.30%          |
>
> Aggregating over all 35,550 questions leads to the same conclusion. The top 25% of models (by competency) contribute on average about **45%** of a question’s total difficulty, whereas the bottom 25% contribute only about **7.9%**. Taken together with the breakdown above for question 14703, this shows that RankLLM’s difficulty estimates are dominated by medium and strong models, while weak models have only a limited effect.
>
> To stress-test the scenario with “too many weak models,” we start from the original pool of 30 real models, select the 10 lowest-scoring ones, and generate 20 synthetic weak models from them, increasing the proportion of the weakest tier from 10/30 to 30/50. We then rerun RankLLM on this 50-model pool with the same normalization scheme (the strongest model scaled to 100, others scaled proportionally). Among the original 30 real models, the **top 20 models’ rankings remain identical** to the original run, and only the bottom 10 models exhibit small shifts of at most two positions. For the top-10 models, the average change in the absolute competency score is only about **0.6 points**. Even under this adversarial construction with heavily duplicated weak models, the scores and rankings of medium and strong models remain **highly stable**, and the effect hypothesized by the reviewer—“adding many weak models substantially boosts some models”—does not appear.
>
>
> This behavior is also aligned with the current LLM ecosystem, where **small models greatly outnumber large models**. Our model pool is intentionally constructed to reflect this reality, with many small and medium-sized models across multiple families, so that the inferred question difficulties reflect weaknesses of the **actual model landscape** rather than an idealized setting defined only by a few top-tier systems.

---

> ### Author Response · Authors · 2025-11-14
> **Response to Weaknesses 2—Score Definition and Cross-Study Comparability**
>
> ## II. Score Definition and Cross-Study Comparability
>
> RankLLM is indeed designed to produce **relative difficulty and competency scores** that are defined with respect to the specific pool of models and questions. Concretely, scores are obtained by **iterative propagation on the model–question bipartite graph** and then **normalized** (e.g., the strongest model is rescaled to 100, and all other scores are interpreted relative to it). As a result, the numerical scale naturally depends on the participating models and the data mix. This behavior is not a special weakness of RankLLM, but a **generic property of comparative, performance-based latent-variable methods** such as IRT, Elo, and Glicko, where ability and difficulty parameters are always pool-dependent and must be re-calibrated when the test or population changes. In practice, these frameworks are widely used because what matters is who is stronger/weaker and which items are harder/easier, not an absolute physical unit of “difficulty”.
>
> Importantly, we show that although the numeric scale is pool-dependent in theory, the **relative structure** that drives conclusions is **highly stable under realistic changes to the pool**. In our robustness experiments, we randomly remove 1/5/10/15 out of 30 models and recompute RankLLM: question-side difficulty rankings remain strongly correlated (0.93–0.99), and model-side competency rankings remain above 0.99 correlation. Similarly, in our dataset-level perturbation experiments, perturbing the question pool leads to consistently high correlations in model rankings. These results indicate that even when the baseline shifts due to pool changes, the **induced ordering of models and the geometry of question difficulty are very robust** and do not fluctuate arbitrarily.
>
> Regarding comparability across model clusters, data clusters, or studies over time, our position is intentionally conservative. Because RankLLM produces pool-dependent relative scores, we do **not** advocate interpreting scores from two disjoint model–question pools as directly comparable. Instead, whenever practitioners wish to compare two models, these models should be **placed in the same evaluation pool** and RankLLM should be run once on that joint pool. This is exactly how we use RankLLM in our experiments: new models are added into an existing pool and evaluated under the same conditions. Our robustness results show that RankLLM’s rankings are highly stable when we add or remove a moderate number of models or perturb the question set, so this protocol yields consistent and interpretable comparisons without requiring an absolute, cross-pool scale.
>
> Overall, the goal of RankLLM is not to introduce an absolute, pool-independent unit of ability or difficulty, but to provide **fine-grained, human-aligned rankings** of models and questions within realistic benchmark settings, and to demonstrate strong alignment with human judgments, robustness, and scalability in that regime. In line with this perspective, our long-term vision is to build a **unified leaderboard** in which a large set of LLMs and benchmark questions are evaluated within a single RankLLM framework. In such a unified setting, most practical comparisons between models are naturally made **within one common model–question pool**, so their competencies and the associated question difficulties are already placed on the same reference scale, and concerns about incompatible score units across studies are largely avoided by design rather than retrofitted through post-hoc linking. In line with this perspective, our **long-term vision is to build a unified leaderboard** in which a large set of LLMs and benchmark questions are evaluated within a single RankLLM framework, similar in spirit to large-scale leaderboards on Hugging Face. In such a unified setting, models are always assessed in a **common model–question pool**, so their competencies and the associated question difficulties are placed on the **same reference scale**, substantially mitigating concerns about incompatible score units across studies.

---

> ### Author Response · Authors · 2025-11-20
> **Follow-up on Response Regarding Stability and Comparability**
>
> Dear Reviewer R6mp,
>
> Thank you again for your time and constructive feedback.
>
> We are writing to follow up on our responses posted on **November 14**. In our reply, we provided a detailed breakdown regarding your primary concerns on **1. ranking stability under weak models** and **2. cross-study comparability**.
>
> We hope our clarification helps to better illustrate the robustness of RankLLM's underlying mechanism. Specifically, we demonstrated that due to the competency-weighted formulation, the influence of weak models is mathematically dampened, preventing the inflation of difficulty scores that you initially worried about. The additional stress tests confirmed that this design allows the framework to maintain stable and reliable rankings for top-tier models, even in the presence of many weak baselines.
>
> As the discussion period progresses, we would greatly appreciate it if you could check whether these explanations and new results have resolved your concerns. If so, we would be grateful if you could **reconsider your rating of our work**. We remain fully available should you have any further questions.
>
> Best regards,
>
> RankLLM Authors

---

### Author Response · Authors · 2025-12-03
**Clarification on Rebuttal-Period Review Updates**

Dear Area Chair,

Thank you for your time and for the extra effort and responsibility you have taken on during this challenging period for the community.

At a high level, RankLLM aims to make LLM evaluation **more informative than flat accuracy** by incorporating **question difficulty** into the ranking signal, which enables **finer-grained and more reliable comparisons**, especially among closely matched models.

Regarding our submission, we would like to clarify that **all concerns raised by the four reviewers have been addressed thoroughly** in our rebuttal and discussion responses. Two reviewers explicitly acknowledged this:

* Reviewer **XEuy** stated that our responses resolved their concerns and that they **“will increase rating to 6”**.
* Reviewer **Kyip** also confirmed that our explanations were satisfactory and that they **“will increase their rating”**.

The remaining two reviewers did not have the opportunity to post follow-up comments or revise their scores due to the unexpected freeze of the discussion period. We believe our rebuttal **fully addressed their concerns** as well, at the same level of completeness as the reviewers who explicitly acknowledged our responses and indicated rating increases. In light of the two reviewers’ clear recognition and score updates, we respectfully ask you to carefully review our responses to the remaining reviewers’ points and make your own judgment based on the resolved concerns reflected in the discussion record.

For clarity, the rating changes are summarized as follows:

| Reviewer          | Original Rating | Stated Updated Rating     |
| ----------------- | --------------- | ------------------------- |
| **R6mp** | 4               | no update posted           |
| **XEuy** | 4               | 6    (explicitly stated)                     |
| **urAu** | 8               | no update posted             |
| **Kyip** | 6               | 8    (explicitly stated)                     |


We respectfully ask you to evaluate our paper based on the actual state of the discussion and the resolved concerns of all reviewers.

Sincerely,

RankLLM Authors

---

### Meta-Review · Area_Chair_HfEv · 2026-01-06

**Summary:**

This paper introduces RankLLM, a novel evaluation framework designed to address a key limitation in current LLM benchmarks: the lack of differentiation in question difficulty. RankLLM jointly quantifies question difficulty and model competency through a principled, bidirectional score propagation mechanism on a model–question bipartite graph. Experiments evaluate 30 models across 35,550 questions from multiple domains, showing that RankLLM outperforms baselines like IRT in fine-grained model discrimination, while being highly efficient and robust.

**Reviewer Concerns:**

Authors provided mathematical clarification and stress tests showing that weak models have limited influence due to competency-weighted formulation. They also acknowledged that RankLLM produces relative scores (similar to IRT/Elo) and is designed for within-pool comparison, with robust rankings under pool perturbations. Moreover, the authors conducted sensitivity analysis for α (rankings remain stable, ρ > 0.99). They explained that family-level blind spots do not dominate due to diverse model pools and dynamic updates. They also rooted the difficulty definition in psychometric theory (IRT), validating it via human alignment.

**Reviewer Scores:**

The final overall score is above the acceptance threshold.

---

### Decision · Program_Chairs · 2026-01-26

Accept (Poster)